# Commissioning a newly developed treatment planning system, VQA Plan, for fast-raster scanning of carbon-ion beams

**Masashi Yagi**[1,2]*, **Toshiro Tsubouchi**[2], **Noriaki Hamatani**[2], **Masaaki Takashina**[2], **Hiroyasu Maruo**[3], **Shinichiro Fujitaka**[4], **Hideaki Nihongi**[5], **Kazuhiko Ogawa**[6], **Tatsuaki Kanai**[2]

1 Department of Carbon Ion Radiotherapy, Osaka University Graduate School of Medicine, Suita-city, Osaka, Japan, 2 Department of Medical Physics, Osaka Heavy Ion Therapy Center, Chuo-ku, Osaka-city, Osaka, Japan, 3 Department of Radiation Technology, Osaka Heavy Ion Therapy Center, Chuo-ku, Osaka-city, Osaka, Japan, 4 Hitachi, Ltd. Research & Development Group, Hitachi-shi, Ibaraki, Japan, 5 Hitachi, Ltd. Smart Life Business Management Division/Healthcare Business Division, KOIL TERRACE 3F 226-44-141-1, Wakashiba, Kashiwa-shi, Chiba, Japan, 6 Department of Radiation Oncology, Osaka University Graduate School of Medicine, Suita-city, Osaka, Japan

* m.yagi@radonc.med.osaka-u.ac.jp

**Data Availability Statement:** All relevant data are within the paper and its Supporting Information files.

## Abstract

In this study, we report our experience in commissioning a commercial treatment planning system (TPS) for fast-raster scanning of carbon-ion beams. This TPS uses an analytical dose calculation algorithm, a pencil-beam model with a triple Gaussian form for the lateral-dose distribution, and a beam splitting algorithm to consider lateral heterogeneity in a medium. We adopted the mixed beam model as the relative biological effectiveness (RBE) model for calculating the RBE values of the scanned carbon-ion beam. To validate the modeled physical dose, we compared the calculations with measurements of various relevant quantities as functions of the field size, range and width of the spread-out Bragg peak (SOBP), and depth–dose and lateral-dose profiles for a 6-mm SOBP in water. To model the biological dose, we compared the RBE calculated with the newly developed TPS to the RBE calculated with a previously validated TPS that is in clinical use and uses the same RBE model concept. We also performed patient-specific measurements to validate the dose model in clinical situations. The physical beam model reproduces the measured absolute dose at the center of the SOBP as a function of field size, range, and SOBP width and reproduces the dose profiles for a 6-mm SOBP in water. However, the profiles calculated for a heterogeneous phantom have some limitations in predicting the carbon-ion-beam dose, although the biological doses agreed well with the values calculated by the validated TPS. Using this dose model for fast-raster scanning, we successfully treated more than 900 patients from October 2018 to October 2020, with an acceptable agreement between the TPS-calculated and measured dose distributions. We conclude that the newly developed TPS can be used clinically with the understanding that it has limited accuracies for heterogeneous media.

**Funding:** This work was partially supported by the JSPS KAKENHI 17K16437. There was no additional external funding received for this study.

**Competing interests:** The authors have declared that no competing interests exist.

## Introduction

Heavy charged particles, particularly carbon ions, have attracted growing attention as cancer therapy modalities worldwide. These ions have good characteristics, including a favorable depth–dose profile, minimal lateral scattering, and increased biological effectiveness around the Bragg peak. Three-dimensional (3D) pencil-beam scanning maximizes these characteristics for carbon-ion therapy (CIT).

The Osaka Heavy Ion Therapy Center (OHITC) in Osaka, Japan—the sixth institute dedicated to CIT in Japan—implemented a 3D pencil-beam scanning system (raster scanning), in which beam delivery does not switch off the scanning beam during the transition time from one spot exposure to the next [1]. The scanning system uses hybrid depth scanning in which the beam range is controlled by inserting acrylonitrile butadiene styrene plates of given thicknesses at selected energies provided by an accelerator. Scanned carbon-ion dose calculation is performed using a treatment planning system (TPS), VQA Plan (Hitachi, Ltd., Tokyo, Japan) [2], which was newly developed for a carbon-ion beam (CIB) at OHITC. The TPS determines the irradiation parameters required to achieve a prescribed dose distribution by accumulating the dose provided by a scanned pencil beam. The determined parameters are sent to the control system of the treatment machine via an electronic medical record system provided by ShadeQuest/TheraRIS (FUJIFILM Medical Solutions Corporation, Tokyo, Japan). The TPS calculates physical doses using an analytical dose calculation algorithm, a pencil-beam [3] model with a triple Gaussian form [4] for the lateral-dose distribution, and a beam splitting algorithm [5] to consider lateral heterogeneity in a medium. To the best of our knowledge, for the first time, a mixed beam model [6] was adopted as the relative biological effectiveness (RBE) model for calculating the RBE of scanned CIBs. These algorithms are used for optimization and clinical dose calculations for all patients treated at our center. They have never been implemented previously in an integrated TPS.

To the best of our knowledge, commercially available TPSs that can calculate exposures of scanned CIBs are inadequate. Monaco-I (Elekta AB, Stockholm, Kingdom of Sweden)—a commercial version of the XiDose developed at the Japan Agency for Quantum and Radiological Science and Technology—has implemented a triple Gaussian trichrome-beam model [7] with a nuclear interaction correction [8] and a pencil-beam redefinition algorithm [4] to consider lateral heterogeneities. This TPS uses the microdosimetric kinetic (MK) model [9] as the RBE model. RayStation (RaySearch Laboratories AB, Stockholm, Kingdom of Sweden) is another TPS used for carbon-ion treatment planning. It uses a fluence dose model as the physical dose model. Either the local effect model I [10] or the MK model can be used as the RBE model. The lateral-heterogeneity correction and biological dose calculation method are the main differences between two TPSs and VQA Plan. In addition, the VQA Plan vendor is the same as the treatment machine other than these two TPSs. The TPS can include the machine control parameters in detail and control the treatment machine accurately.

In an inhomogeneous geometry, the pencil-beam model only calculates the water-equivalent range along the axis of the scanning pencil beam; off-axis heterogeneities are not considered in the calculation of the dose deposited by the scanned pencil beam. To consider lateral heterogeneity in a medium, the TPS implements a beam splitting algorithm in phase space [5]. The algorithm uses the self-similarity of Gaussian distributions, allowing Gaussian beams to be split into narrower and deflected sub-beams.

Single-energy kilovolt X-ray computed tomography (CT) is commonly used in carbon-ion treatment planning to calculate the dose distribution. The water-equivalent depth used in the pencil-beam model is calculated using effective-depth computations, with corrections for longitudinal heterogeneity. The effective depth is a linear integration of the stopping power

relative to water along the beam path using the CT number-to-relative-stopping-power (RSP) curve. Hence, accurate calibration of the CT number-to-RSP curve is required to determine the carbon-ion range and make accurate dose calculations, especially in an inhomogeneous geometry. Unlike photon radiotherapy [11], for carbon-ion radiotherapy, the measured correlation between the CT number and RSP for various body tissues is insufficient to enable real tissues to be represented with available artificial materials [12]. Among several methods available for performing the calibration [13–15], we adopted a theoretical approach—stoichiometric calibration—for the commissioning work.

In CIT-treatment planning, a clinically relevant dose—a biological dose defined as the product of the absorbed dose and RBE—is optimized for each clinical case. The RBE depends on various factors, such as linear energy transfer (LET), particle species, track structure, dose, dose rate, oxygen pressure, endpoint, and tissue type. In Japan, the survival of human salivary gland (HSG) tumor cells has been used as the endpoint for determining the RBE of CIBs, assuming a normal oxygen pressure and irradiation with a normal dose rate, and it is used for all types of tumors and normal tissues [6, 9, 16]. Notably, HSG tumor cells are assumed to be representatives of cells responding to CIBs, and the response to individual tumors is considered based on the HSG response. We adopted the mixed beam model in the TPS because this model has been used in most published clinical results for CIBs [17] in Japan. Thus, the accumulated evidence, especially for the dose tolerances of normal organs, can be easily referenced using this model to estimate clinical outcomes for specific tumors.

## Materials and methods

### Raster-scanning beam delivery system

The characteristics of the treatment machine will be reported separately; therefore, we only briefly describe it. The system uses raster scanning in which the beam is not switched off between the exposed spots. A hybrid scanning system that can provide 100 different nominal energy selections between 73.3 and 430.0 MeV/u from 12 accelerated energies, corresponding to carbon-ion ranges of 0.5–30.2 $g/cm^2$ in water, is available from the synchrotron (HIMAK, Heavy Ion Medical Accelerator in Kansai) in the OHITC (HyBeat Heavy-ion Therapy System, Hitachi, Ltd., Tokyo, Japan). The spot depth is controlled in 3-mm increments via the hybrid depth-scanning techniques; it is not adjustable in arbitrary increments. The field size can be set up to $200 \times 200$ $mm^2$ at the isocenter. After entering the nozzle, a pencil beam first goes through an air-filled dose monitor. The main dose monitor checks the dose remaining at a given position, and a spot position monitor (SPM) filled with an $Ar$-$CO_2$ mixture confirms the spot position for every energy. A monitor unit (MU) is defined based on a fixed amount of charge collected in the ionization chamber of the main dose monitor, and it corresponds to a physical dose to the water of 1-cGy defined in a volume of $100 \times 100 \times 100$ $mm^3$, with 200 mm in range yielding irradiation of 200 MU for 200 cGy. The minimum and maximum values that can be delivered at each spot are 0.0006 and 0.15 MU, respectively. The resolution is 0.00001 MU. The dose rate can be changed from 1.0 to 8.0 MU/s with a 0.1-MU/s resolution.

### Commissioning the treatment planning system

The TPS used in this study is VQA Plan version 5.8 (Hitachi, Ltd., Tokyo, Japan), which includes a carbon module for scanned CIB delivery. Beam modeling comprises two steps: physical and biological dose modeling. The TPS uses an analytical dose calculation algorithm and a pencil-beam model with a triple Gaussian form for the lateral-dose distribution. Input data required for the scanned CIB are integral depth doses (IDDs), which are measured using StingRay (IBA Dosimetry GmbH, Schwarzenbruck, Germany), in-air lateral profiles measured

using a PinPoint 3D chamber (type 31016, PTW-Freiburg GmbH, Freiburg, Germany), measurements relating to fragmentation acquired using the PinPoint chamber, the absolute dose obtained using an Advanced Markus chamber (AMC) (type 34045, PTW-Freiburg GmbH, Freiburg, Germany), and the absolute dose correction factor. Moreover, the TPS requires beam data obtained with a range shifter (RS). Modeling the biological dose requires the dose-averaged LET ($LET_d$) and linear–quadratic model (LQM) parameters (i.e., $\alpha$ and $\beta$, respectively, which are the coefficients of the LQM). Finally, a scaling factor is essential to convert biological doses into clinical doses.

## VQA beam modeling

Details of the method used to model the physical and biological doses from the measurement data are described in [2].

## Physical dose

The physical dose is calculated using an analytical dose calculation algorithm and a pencil-beam model with a triple Gaussian form for the lateral-dose distribution. Let the central axis of the beam be the $z$-axis, with the plane of the isocenter defined to be $z = 0$ mm and the positive $z$-direction toward the carbon-ion source, and let the $x$- and $y$-axes be the transverse coordinates. The dose deposited in each spot is calculated at each point of the calculation grid by adjusting the water-equivalent distance to that point. A general 3D dose at the calculation point (or voxel) $d_i$ can be written as follows:

$$d_i(x_i, y_i, z_i) = \sum_j w_j \sum_{n=1}^{3} f_j^n(z_i) G_j^n(z_i), \tag{1}$$

where $w_j$ is the number of particles in CIB $j$ in MU, $f_j^n(z_i)$ denotes the IDD of the $n$-th component, and $G_j^n(z_i)$ is a two-dimensional (2D) lateral Gaussian distribution at depth $z_i$.

$$G_j^n(z_i) = \frac{1}{2\pi\sigma_{n,x}(z_i)\sigma_{n,y}(z_i)} exp\left[-\frac{(x_i - x_j(z_i))^2}{2\sigma_{n,x}(z_i)^2}\right] exp\left[-\frac{(y_i - y_j(z_i))^2}{2\sigma_{n,y}(z_i)^2}\right], \tag{2}$$

where $\sigma_{n,x}$ and $\sigma_{n,y}$ are the beam sizes in the $x$- and $y$-directions, respectively; at depth $z_i$, $x_i$ is the lateral position of point $i$ and $x_j$ is the center position of beam $j$ at depth $z_i$. In Eqs (1) and (2), the first component ($n = 1$) represents the incident CIB and the beam sizes in the $x$- and $y$-directions are calculated by considering the optical parameters of the beam and multiple Coulomb scattering in a medium. The second and third components ($n = 2, 3$) correspond to fragments with small- and large-angle scatterings, respectively, and the beam sizes are isotropic in the $x$- and $y$-directions in this beam model. The quantities $f_j^n(z_i)$ and beam sizes ($\sigma_{n,x}$ and $\sigma_{n,y}$) in the lateral directions are parameters to be determined in the physical beam-modeling process.

A beam splitting algorithm [5] in phase space is applied to the first component to enable appropriate consideration of lateral heterogeneity in the medium. The second and third components are calculated by considering density information only along the scanning beam's central axis. The first component is split into nine sub-beams, with the beam ellipse in phase space split into three sub-beams in both the $x$- and $y$-directions; this splitting number is not adjustable by the user. The beam splitting is performed only once at 700-mm upstream from the isocenter plane using the registered optical parameters; this differs from the method described in [5].

Considering the transit dose obtained during raster scanning, the dose delivered while the spot is moving is approximated as a spot dose delivered at a given time along the direction of motion from one spot to the next. This calculation considers the scanning speed in each scanning direction, and the amount is calculated by multiplying the beam current defined at every control point by the period.

A control point is created for the following during optimization: 1) a short dose used for planning dose irradiation; 2) a change of range (i.e., for a different accelerated energy); 3) if the time required to stop the beam is shorter than 90 μs; 4) if the amount of the dose needed to calculate the beam width in the SPM is less than 0.00018 MU; and 5) if the ratio of the stopping dose, which is the deposited dose at a point, and the moving dose, which is the transit dose, is not 1:1.

## Biological dose

Empirically, carbon-ion irradiation is more effective biologically than photon irradiation; therefore, the RBE has to be calculated using a bio-mathematical model to obtain the prescribed dose. We adopted the mixed beam model—also called the Kanai model—as the RBE model for the biological dose calculation. In this model, the RBE is calculated from the cell survival curves for photons and carbon ions using a specified survival level (10%) based on the LQM with HSG. Thus, the biological dose is given by

$$d_{bio,i} = \frac{\sqrt{\alpha_X - 4\beta_X \ln(S_i)} - \alpha_X}{2\beta_X}, \tag{3}$$

where $\alpha_X$ and $\beta_X$ are the LQM parameters for photons. $S_i$, which describes the cell survival at voxel $i$, is represented as

$$S_i = exp(-\alpha_i d_i - \beta_i d_i^2), \tag{4}$$

where $\alpha_i$ and $\beta_i$ are the LQM parameters for carbon ions. The effective survival fraction for a specific particle spectrum is produced by the contributions of all monoenergetic beams to provide the total dose $d$ at voxel $i$.

The employed TPS applies an improved mixed beam model [18] in which LQM parameters are divided into two ion types. LQM parameters relating to the carbon ions are applied to carbon isotopes, whereas those for helium are used for fragment isotopes other than carbon.

$$\alpha_i = \frac{1}{d_i} \sum_j w_j (\alpha_{ij}^{(C)}(LET_C) d_{ij}^{(C)} + \alpha_{ij}^{(He)}(LET_{frag}) d_{ij}^{(frag)}), \tag{5}$$

$$\sqrt{\beta_i} = \frac{1}{d_i} \sum_j w_j (\sqrt{\beta_{ij}^{(C)}(LET_C)} d_{ij}^{(C)} + \sqrt{\beta_{ij}^{(He)}(LET_{frag})} d_{ij}^{(frag)}), \tag{6}$$

where $\alpha_i^{(C)}$ and $\beta_i^{(C)}$ are LQM parameters for the carbon isotopes and $\alpha_i^{(He)}$ and $\beta_i^{(He)}$ are the LQM parameters for fragment isotopes other than carbon. The carbon and fragment isotope doses are $d_{ij}^{(C)}$ and $d_{ij}^{(frag)}$, respectively. These LQM parameters are functions of the LET and are obtained from [18]. As the biological effect in a cell is determined by the total effect of the dose contributions from different LET components that comprise the radiation field, we used $LET_d$ in the calculations. The $LET_d$ data for each scanning beam were obtained from Monte Carlo simulations (MCSs) after tuning the physical parameters of the simulation to express the actual beam.

To consider the biological effects of the two types of ions separately, the physical dose provided by Eq (1) can be divided into two terms, which represent the dose contributions received

from the carbon and fragment isotopes. In the triple Gaussian form, the first component ($n = 1$) represents the carbon isotopes, the second component ($n = 2$) is assumed to be a mixture of carbon and fragment isotopes, and the third component ($n = 3$) corresponds to the fragment isotopes. Hence, the physical dose contributions of the carbon and fragment isotopes are related to the triple Gaussian components as follows:

$$
\begin{aligned}
d_i(x_i, y_i, z_i) &= \sum_j w_j (d_{ij}^{(C)} + d_{ij}^{(frag)}) \\
&= \sum_j w_j [(f_j^1(z_i) G_j^1(z_i) + (1 - R(z_i)) f_j^2(z_i) G_j^2(z_i))^{(C)} \\
&\quad + (R(z_i) f_j^2(z_i) G_j^2(z_i) + f_j^3(z_i) G_j^3(z_i))^{(frag)}],
\end{aligned}
\tag{7}
$$

where $R(z_i)$ is the ratio of the fragment isotopes in the range of atomic numbers from $Z = 1$–5 to all fragment isotopes, which is obtained from the MCSs.

We established the RBE model based on results from preclinical studies. However, the RBE value has to be related to clinical data. For this purpose, we introduce the concept of a "clinical RBE." In the mixed beam model, the clinical dose is provided by

$$
d_{clin,i} = 1.46 \times d_{bio,i},
\tag{8}
$$

where $d_{clin,i}$ denotes the clinical dose at voxel $i$. The scaling factor was 1.46, which was determined by connecting the modeled RBE to the clinical experience with neutron therapy [16].

## Required input data

Modeling of scanned CIB requires IDDs, in-air lateral profiles, measurements to determine the fragment components, the absolute dose, and the absolute dose correction factor. The TPS also requires beam data obtained with an RS.

In the beam-modeling process, these input data were used to tune physical parameters applied in the MCSs and determine the analytical parameters needed in the calculation algorithm. Particularly, the IDDs were used for tuning Geant4.9.3 [19]. We used the in-air lateral profiles to determine the optical and scattering-power parameters for multiple Coulomb scattering in the first component of the CIB. We utilized the fragment component measurements to determine the parameters that describe the ratio between the second and third beam components and the beam sizes of each component. Further, we used the absolute dose measurements to convert the IDDs to absolute values. The absolute correction factor was used to compensate for the difference between the calculated physical absolute doses and those obtained from the measurements.

## Determination of the CT number-to-RSP curve, validation, and range error estimation

Uncertainties in the CT number-to-RSP influence the dose calculation accuracy. For patient simulation, we used an Aquilion LB CT (Canon Medical Systems Corporation, Tokyo, Japan) scanner. We adopted stoichiometric calibration [12] to determine the CT number-to-RSP curve. Using MODEL 062M (CIRS, Inc., Norfolk, VA, USA), we examined various CT acquisition settings and phantom sizes to establish an imaging protocol and create a CT number-to-RSP table, which the TPS uses to estimate the RSP ratios for CIB dose calculations. The compositions of the inserts in the phantom, which are used for CT number calibration, were provided by the vendor. The tissues used to calculate the RSP values were adopted from the representative tissues described in [13], which are based on publication 110 of the

International Commission on Radiological Protection (ICRP) [20]. These tissues differ from those used in [12]. The compositions and physical densities of 11 representative tissues are provided in [13]. The mean excitation energy for water was 75.3 eV [21], and the particle velocity relative to the speed of light in a vacuum was 0.481 (131.0 MeV/u) [22]. We obtained the mean excitation energy for the representative tissues. We calculated the RSP using the Bethe–Bloch equation without any energy-dependent corrections.

Beyond the calibrated range, the TPS treats the RSP as a constant, which is either the minimum or maximum RSP value. The RSP value of gold, determined in the same manner as the stoichiometric calibration, is the highest RSP registered; moreover, values lower than the calibrated RSP values were not registered.

A polybinary tissue model [14], which has been used as a standard calibration method for CIT in Japan [23], was also studied independently to verify the RSP calculation using a dedicated phantom (ACTP01 and ACTP02, Accelerator Engineering Corporation, Chiba, Japan). The details of the method can be found in [14]. Briefly, the polybinary tissue model approximates body tissues as mixtures of muscle, air, fat, and bone, for which the surrogates are water, air, ethanol, and a potassium phosphate solution, respectively. After correcting the difference in RSP between the body tissues and tissue substitutes, the polybinary tissue model uniquely determines the CT number–RSP relationship using CT calibrations of four tissue substitutes.

We investigated the calibration table's accuracy using milk, lard, lean meat, and marbled meat as test materials. We placed each test material in a special container (Kyoto Kagaku Co., Ltd., Kyoto, Japan), with the size matching the hole in the MODEL 062M phantom, and inserted them into the phantom center. We measured the CT number using the same scanning condition as that applied for the calibration. We measured the RSP value of each material in the same manner as in [24]. We measured the range of a 302.1 MeV/u CIB using a StingRay and an accordion-type water phantom (AVWP03, Accelerator Engineering Corporation, Chiba, Japan).

The range uncertainty is of major interest for CIT. Therefore, we determined the range uncertainty in our stoichiometric calibration. The details of the method can be found in [25, 26]. In summary, the uncertainties in patient's CT imaging, uncertainties in the parameterized stoichiometric formula used to calculate the theoretical CT number, deviation of actual human body tissue from the standard tissue defined by the International Commission on Radiation Units and Measurements, deviations in the mean excitation energies, and uncertainties due to the energy dependence of the RSP, which are not considered by the dose algorithm, were estimated for lung, soft tissue, and bone. We calculated the composite range uncertainty, $\sigma_R$, by combining the uncertainties estimated for each tissue as follows:

$$\sigma_R = w_L \times \sigma_L + w_S \times \sigma_S + w_B \times \sigma_B, \tag{9}$$

where $w_L$, $w_S$, and $w_B$ are the relative weights of the lung, soft tissue, and bone, respectively, along a ray from the skin to the proximal or distal end of the planning target volume, and $\sigma_L$, $\sigma_S$, and $\sigma_B$ are the corresponding uncertainties in the RSPs.

## Treatment planning system validation

We measured IDDs for pencil beams point by point, with step sizes of 0.2–10 mm, using the StingRay, except for nominal energies below 100 MeV/u. For the nominal energies, we used a field of $100 \times 100$ mm$^2$ instead of a pencil beam and based the IDDs on the dose–area product (DAP) [27] obtained using the AMC set up at the field center. We applied this approach because the wall of StingRay was too thick—4.9-mm water-equivalent length—to allow measurements of a short-range profile. The data acquisition system included a water phantom

(MP3-M, PTW-Freiburg GmbH, Freiburg, Germany) and a dual-channel electrometer (TANDEM, PTW-Freiburg GmbH, Freiburg, Germany). For scanning with a detector, we used the electrometer's trigger mode to acquire a beam-on signal from the accelerator. We aligned the surface of the phantom to the isocenter using a room laser. We acquired in-air profiles with a 2D scintillator (XRV-2000 Falcon Beam Profiler, Logos Systems Int'l, CA, USA). We also measured selected lateral profiles in water point by point, with step sizes of 0.2–1.0 mm, using the PinPoint chamber. To verify the fragment isotopes, we performed frame pattern irradiation using flat concentric squared frames arranged in three selected frame patterns (denoted as A, B, and C). The inner and outer side lengths of frames A, B, and C were 0 and 24 mm, 6 and 18 mm, and 18 and 30 mm, respectively. The PinPoint chamber was placed at a frame's center, and we conducted measurements at several depths. We measured the absolute dose in a $200 \times 200$ mm$^2$ field created by superimposing pencil beams with the AMC at a depth of 20 mm from the surface, except that measurements for 100 and 140 MeV/u were made at 2- and 5-mm depths, respectively. We calculated the DAPs for comparison. Verification of the absolute dose correction factor included measurements of a point dose at the field center and in the spread-out Bragg peak (SOBP) in selected volumetric dose distributions as a function of field size ranging from $20 \times 20$ mm$^2$ to $200 \times 200$ mm$^2$. We created all fields by superimposing pencil beams. We obtained measurements with the AMC, except that we used the PinPoint chamber for field sizes less than $40 \times 40$ mm$^2$ because the AMC was too large for the fields. We aligned the isocenter to the detector using a room laser for this measurement. Moreover, we measured a field size of $100 \times 100$ mm$^2$ via a 6-mm SOBP point by point, with step sizes of 1.0–20 mm, using the PinPoint chamber.

We converted the irradiation pattern files—called the control point file and spot position file—used in the measurements using in-house software created using Python (version 3.6.10) to import them to the TPS. We examined the differences between the measurements and calculations to evaluate the dosimetric parameters ($R_{90}$, the depth defined at 90% of the dose level, the widths at 95%, 50%, and 5% of the dose level, and the penumbra, defined as 20%–80% of the dose level) using the in-house software created with Python. The calculation grid was fixed at $2.0 \times 2.0 \times 2.0$ mm$^3$ and was not adjustable by the user.

## Validation in heterogeneous media

An order-made phantom (Kyoto Kagaku Co., Ltd., Kyoto, Japan) comprising $150 \times 150 \times 20$ mm$^3$ of acrylic and $30 \times 30 \times 20$ mm$^3$ of the lung- and bone-equivalent materials were placed 40 mm upstream from the surface of the water phantom, which we aligned to the isocenter using a room laser. We performed volume irradiations of a $100 \times 100$ mm$^2$ field in the range of 120 mm with a 40-mm SOBP. We used the PinPoint chamber connected to the data acquisition system to measure point doses along the central axis and measure the lateral profiles at selected depths.

We used the in-house software to convert the irradiation pattern files used in the measurements and import them to the TPS. We examined the differences between the measurements and calculations to evaluate the dosimetric parameters ($R_{90}$, the width at 95%, 50%, and 5% of the dose level, and the penumbra) using the in-house software created using Python. In addition, we compared the measurements and calculations using a one-dimensional (1D) local-gamma-index analysis at a distance-to-agreement of 2 mm and a dose difference of 2%, with a 10% threshold. For this analysis, we also used the in-house software created with Python. We measured the RSP for each material comprising the phantom in the same manner as in the experiment used to validate the CT number-to-RSP curve, and this was reflected in the calculation by overwriting the CT number function after inversely estimating the CT number of the

measured RSP in the created CT number-to-RSP table because the material comprising the phantom was not a real tissue and the CT number acquired from the CT image was not as accurate as that of the real tissue.

### Validation of the biological dose calculation

We developed the RBE model for calculating the biological dose based on *in vitro* data. One method to validate the RBE model itself is to compare the calculated RBE values with RBE values measured in a preclinical study. We validated the RBE calculated in the TPS using the RBE measurements reported in [2], and they agreed to be within 10%.

In commissioning the TPS, the biological dose calculation validation determines the calculation accuracy. One validation method used for this purpose is comparing the RBE values calculated in the newly developed TPS with the RBE values calculated in a validated TPS that implements the same RBE model concept as the newly developed TPS. As such, we selected the TPS Xio-N (Elekta AB, Stockholm, Kingdom of Sweden and Mitsubishi Electric Corporation, Tokyo, Japan) as the validated TPS. The biological dose calculation accuracy in Xio-N has been determined at the Gunma University Heavy Ion Medical Center (GHMC) [28].

We performed comparisons for a given volumetric dose distribution with a field size of $60 \times 60$ mm$^2$ and a range of 150 mm as a function of SOBP size ranging from 30 to 120 mm. The prescribed dose was 3.6 Gy (RBE) at the SOBP's center.

### Measurements for patient-specific quality assurance

We performed patient-specific quality assurance (PSQA) to confirm that the treatment system works correctly by comparing the measured dose distribution with the dose distribution calculated in the TPS. All patient records used in this study were approved by OHITC's institutional review board (IRB, No. 20043). A consent form was waived because this was a retrospective study using de-identified data of patients who have completed CIT.

In PSQA, the planned dose distribution calculated with a single-field uniform dose (SFUD), which is designed to produce a uniform dose over the entire target volume for each field, is converted to the dose distribution in the water phantom. In actual measurements, the coincidence between the isocenter established with a room laser and CIB is crucial because the water phantom is aligned to the room laser. This coincidence is guaranteed in daily quality assurance (QA) using an order-made dedicated alignment phantom (Taisei Medical Inc., Osaka, Japan). We performed the measurements using the PinPoint chamber and a commercial 2D ionization chamber array (OCTAVIUS Detector 1500 XDR, PTW-Freiburg GmbH, Freiburg, Germany) mounted on an accordion-type water phantom that enables the water-equivalent thickness to be changed easily [29]. We used a dedicated holder (Accelerator Engineering Corporation, Chiba, Japan) to attach the PinPoint chamber to the accordion-type water phantom. The phantom shortens successive measurements at different depths by driving the motor control remotely. We used the point dose measured with the PinPoint chamber to normalize the point dose at the 2D ionization chamber array's center. We compared the measured dose distributions to the planned dose using a 2D local-gamma-index analysis [30] with the criterion of an acceptance level greater than 90% at a distance-to-agreement of 3 mm and a dose difference of 3% [29], with a 10% threshold. For a detailed analysis, gamma analysis using 2 mm/3%, 2 mm/2%, and 1 mm/1%, with a 10% threshold, were performed. For these analyses, we used VeriSoft version 7.1 (PTW, Freiburg, Germany).

In this study, we analyzed 20 ports from 10 prostate cancer (PC) cases and 23 ports from 6 head and neck cancer (HNC) cases treated at OHITC. We performed the measurements at the

SOBP's center in the PC cases, whereas we performed them at different depths within the SOBP in the HNC cases.

## Results

### Determination and validation of the CT number-to-RSP curve

Fig 1(A) shows the determined CT number-to-RSP curve. The solid line is for small-diameter cases, such as the head and neck region, whereas the dashed line is for large-diameter cases, such as the abdominal region. In regions with high CT numbers, the RSP of the body was lower than that of the head. We observed an ~6% difference in the RSP around the region of 2600 HU, whereas the differences in soft tissue regions were within 1%.

Fig 1(B) and 1(C) depict the comparisons of our curves and curves derived from the poly-binary tissue model. The difference increased gradually as the CT number increased, and the maximum differences were found to occur in the bone mineral area (~−3.0% and −4.7% for the head and body, respectively).

Fig 1(D) and 1(E) show the results of our experimental verifications. The CT numbers of materials used in the experiment varied from ~−100 to +65 HU. The differences between the RSP values determined from the calibrated curves and the experiments were within 1% for both cases.

In our stoichiometric calibration, the uncertainties in the RSP for lung, soft tissue, and bone were found to be 6.6%, 1.4%, and 2.2%, respectively. We also confirmed a range margin of 3.5%, which was sufficient to include the range uncertainty in a composite range calculation.

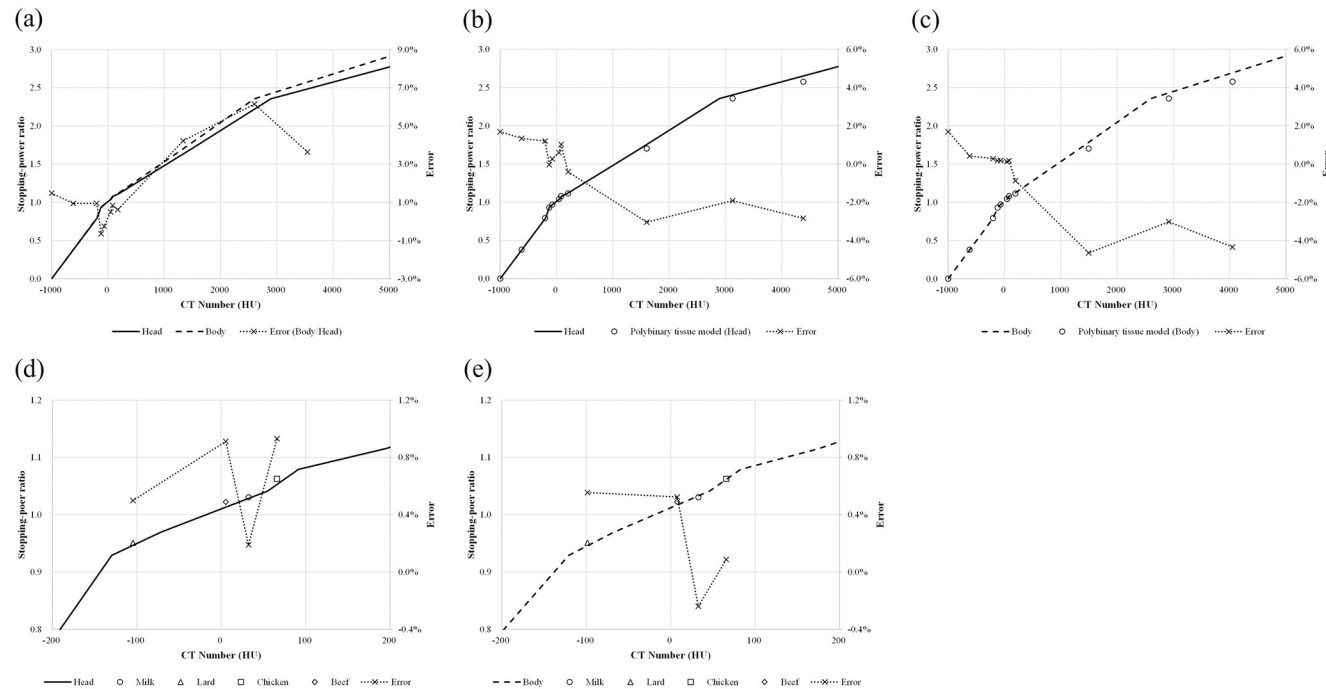

**Fig 1. CT number-to-RSP curve created with stoichiometric calibration.** (a) Relationship between head and body cancer cases—solid lines: head case; dashed lines: body case; dotted lines with crosses: errors. Comparison of curves obtained using our method and using a polybinary tissue model for (b) head and (c) body; the circles show the polybinary tissue model. Comparison between the calculated stopping-power ratio and those measured for materials representing (d) head and (e) body—circles: milk; triangles: lard; squares: chicken; diamonds: beef.

### Treatment planning system validation

We compared the TPS-calculated spot sizes and profiles against data measured at selected energies. Fig 2(A) shows a comparison of the TPS-calculated and measured spot sizes at all energies; they all agreed to be within 0.5 mm in both the *x*- and *y*-directions. The *y*-direction represents the nozzle rotation axis, and the *x*-direction is orthogonal to the *y*-direction. The TPS-calculated in-air profiles agreed well with those measured at representative energies and positions [Fig 2(B) and 2(D)].

The TPS-calculated IDDs agree well with those measured at all energies, including a case with the RS inserted (Fig 3). The distance-to-agreement between the TPS-calculated and measured ranges ($R_{90}$) was within 0.9 mm for all measured pristine Bragg peaks. With no RS, the values of the IDDs at the Bragg peak increased with decreasing carbon-ion energy; however, they decreased after reaching a maximum at 208.3 MeV/u. The dose in the fragmentation tail region increased with carbon-ion energy.

Fig 4(A)–4(F) show the lateral profiles in water for representative energies and positions. The TPS-calculated and measured spot sigmas all agree to be within 0.4 mm with and without the RS, although the tail regions in the profiles show some differences.

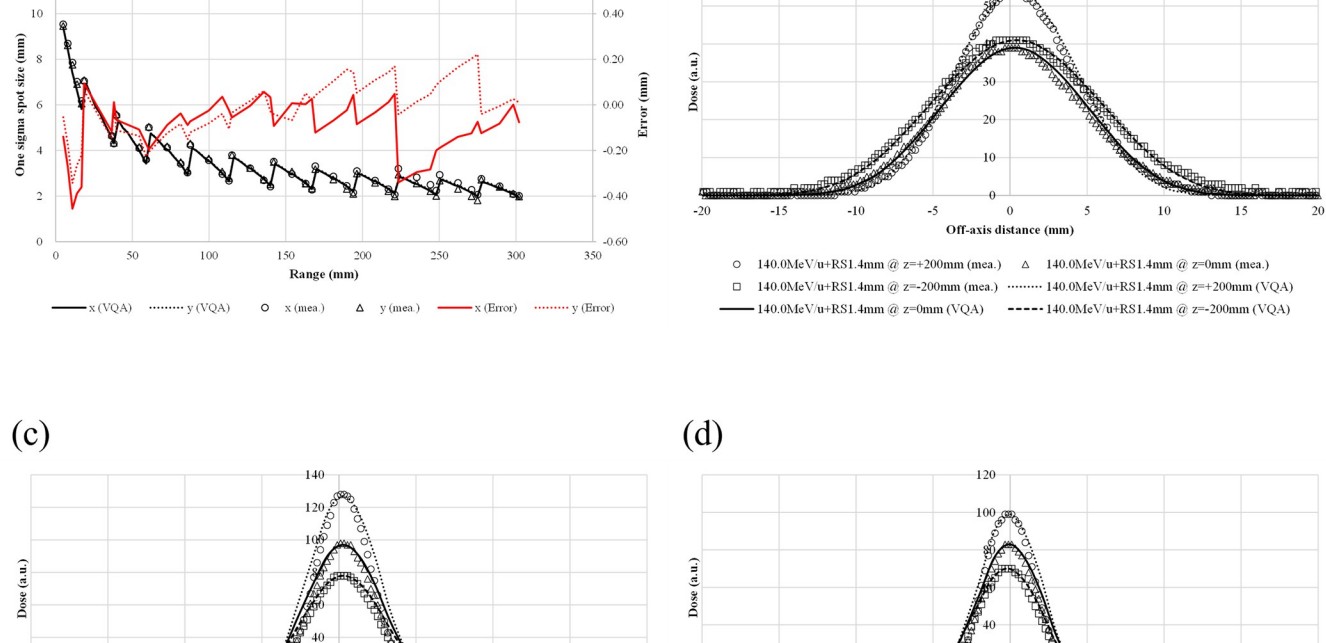

**Fig 2. In-air spot sizes and profiles for selected energies.** (a) The beam size defined by one sigma in the isocenter plane (*z* = 0) vs. energy—solid line: calculated spot size in the *x*-direction; dotted line: calculated spot size in the *y*-direction; circles: measured spot size in the *x*-direction; triangles: calculated spot size in the *y*-direction; red solid line: error in the spot size in the *x*-direction; red dotted line: error in the spot size in the *y*-direction. The lateral in-air dose profiles are shown for a pencil beam at selected energies and different positions. The selected energies are (b) 140.0 MeV/u, (c) 302.1 MeV/u, and (d) 430.0 MeV/u. The lateral in-air dose profiles are measured at *z* = +200 mm (circles), 0 mm (triangles), and −200 mm (squares), whereas they are calculated at *z* = +200 mm (dotted lines), 0 mm (solid lines), and −200 mm (dashed lines).

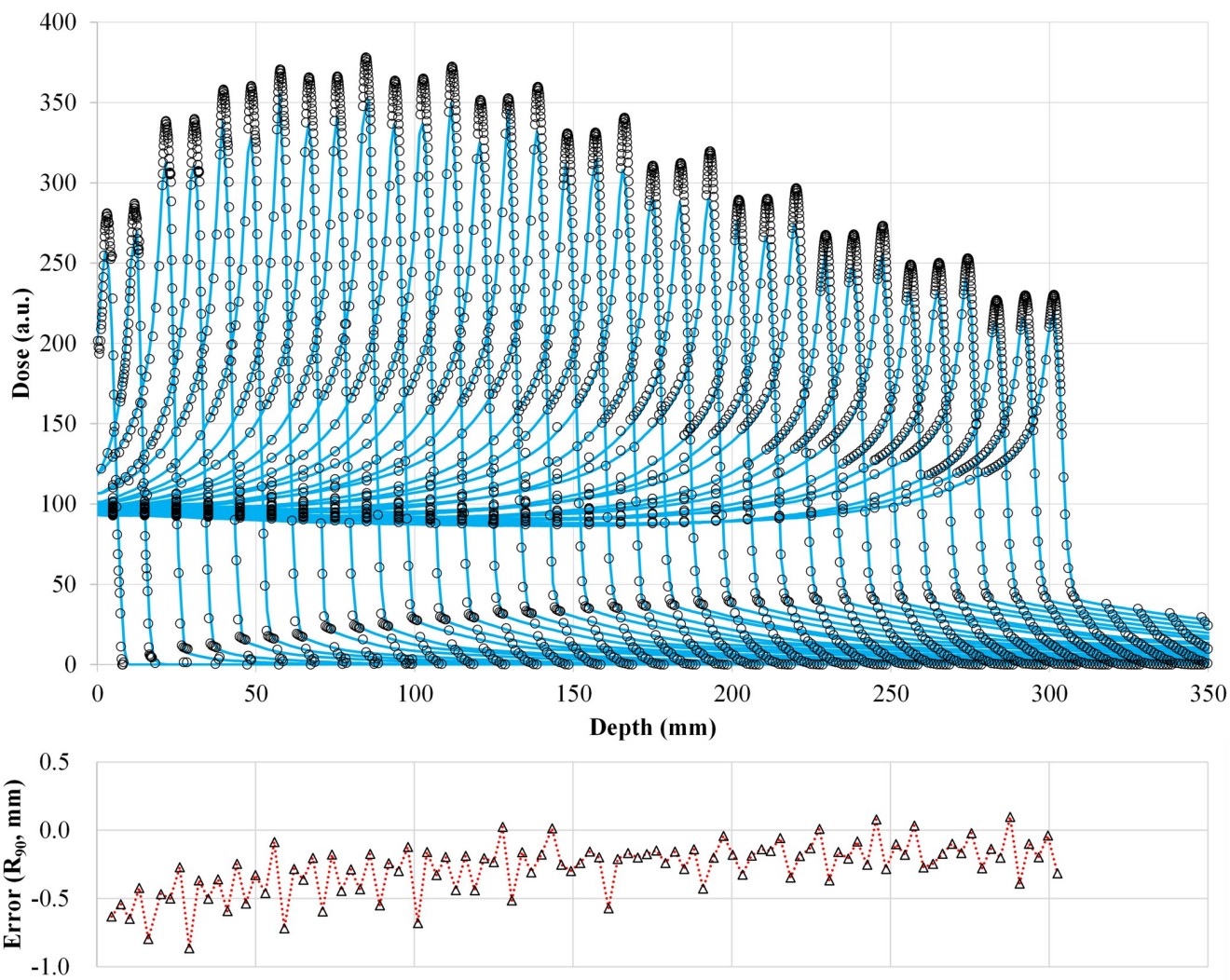

**Fig 3. Comparison of IDDs from the TPS calculations and measurements for 100 energies.** The circles and solid lines show the measured and calculated IDDs, respectively. IDDs were selected for ease of viewing. The lower graph shows the differences in $R_{90}$ in units of millimeters for all energies.

Fig 5 shows a comparison between the TPS-calculated and measured point doses. The maximum difference in the absolute dose between the calculation and measurement was 1.8% for energies above 140.0 MeV/u and −6.6% at 100.0 MeV/u with the RS.

The TPS-calculated and measured point doses as functions of depth in the irradiation pattern are shown in Fig 6(A)–6(F). The TPS-calculated point dose deviated from the measured values by −3.7% to +0.5%, −2.8% to +6.8%, and 11.9% to 39.3% for patterns A–C, respectively, in a 430.0-MeV/u CIB without RS. The deviation increased as the length of the inner side increased (i.e., from A to C), independent of the carbon-ion energy.

Fig 7(A)–7(C) show the absolute dose correction for volume irradiation. The difference between the TPS-calculated and measured doses reached a maximum at the field size of $200 \times 200$ mm$^2$ for all examined conditions. The maximum difference between the TPS-calculated and measured doses was −2.4%.

The TPS-calculated depth and lateral-dose profiles in a 6-mm SOBP are compared with those obtained from the measurements (Fig 8). The difference between the TPS-calculated and

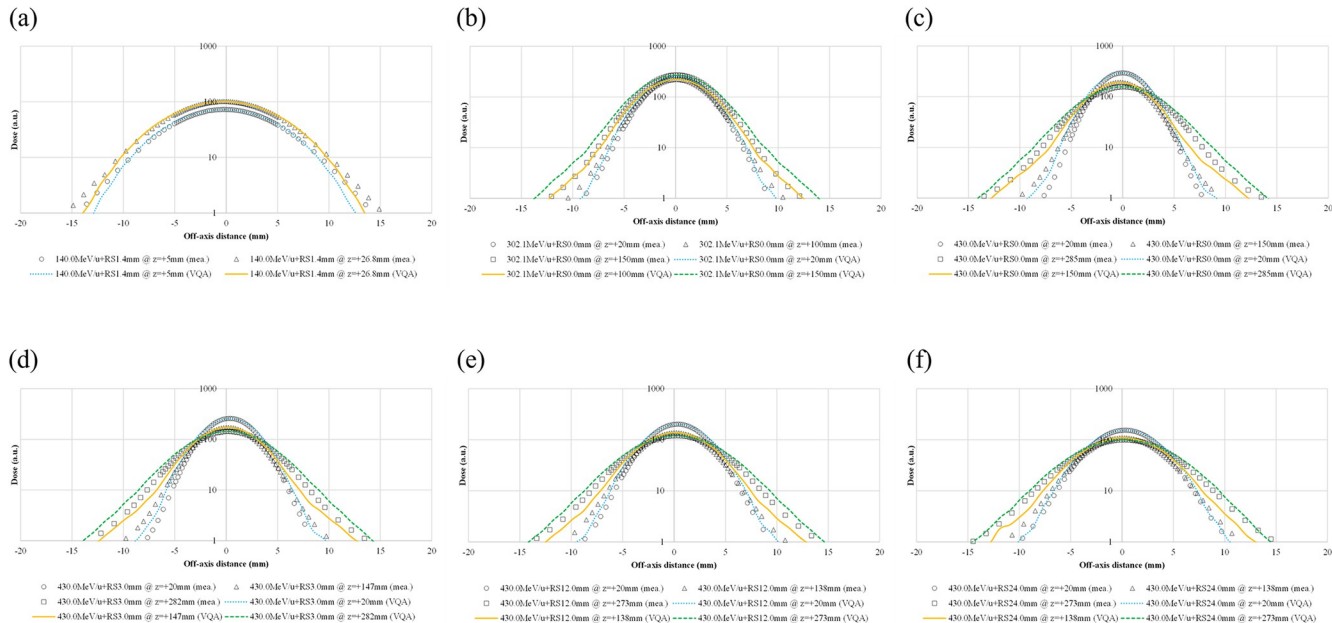

**Fig 4. Lateral-dose profiles in water at selected energies and different positions.** The profiles are for pencil beams with energies of 140.0, 302.1, and 430.0 MeV/u (a–c, respectively). Pencil-beam profiles at an energy of 430.0 MeV/u for cases with RS thicknesses of 3.0, 12.0, and 24.0 mm (d–f, respectively).

measured $R_{90}$ was within 0.9 mm. The shape of the lateral profile at 156 mm was close to the measured one, showing that the difference in the penumbral width was less than 0.5 mm. The width of the dose profile was −1.8, −0.5, and 0.0 mm at 95%, 50%, and 5% of the examined dose level, respectively.

## Validation in heterogeneous media

We evaluated the lateral-heterogeneity correction accuracy by performing measurements involving heterogeneous conditions. Here, we show only the results for a case with nine sub-beams (3 × 3) because this is the default number in the TPS. Cases with different beam splitting numbers are considered in the discussion.

Fig 9(A) shows the heterogeneous phantom and measurement setup. The measured RSP values for the acrylic, lung-, and bone-equivalent materials used in the analysis were predetermined to be 1.16, 0.11, and 1.37, respectively, whereas the TPS-calculated RSP values for these materials were 1.16, 0.14, and 1.38, respectively. As shown in Fig 9, the TPS-calculated dose was smaller than the measured dose, regardless of the materials. The maximum difference between the TPS-calculated and measured dose was observed to be −1.0%, −3.9%, and −5.6% in the acrylic, lung-, and bone-equivalent materials, respectively. The TPS-calculated range was shorter than the measured range, regardless of the materials. For the acrylic material, the range was measured at the phantom's center (i.e., at 0.0 mm in the off-axis distance). The differences in $R_{90}$ for the acrylic, lung-, and bone-equivalent materials were −2.2, −1.2, and −0.9 mm, respectively. The dose in the fragmentation region at the center of the acrylic material at 105 mm exhibited a larger difference—up to 34.6% corresponding to 1.73 of gamma index [Fig 9(B) and Table 1]. The shape of the lateral profile at 66 mm was close to the one measured, showing that the difference in the penumbral width was less than 0.3 mm, and the difference in the dose profile width was less than 1.5 mm at the examined dose level [Fig 9(C) and Table 1]. The shape at 105 mm differed in the lung-equivalent material region; the shoulder of

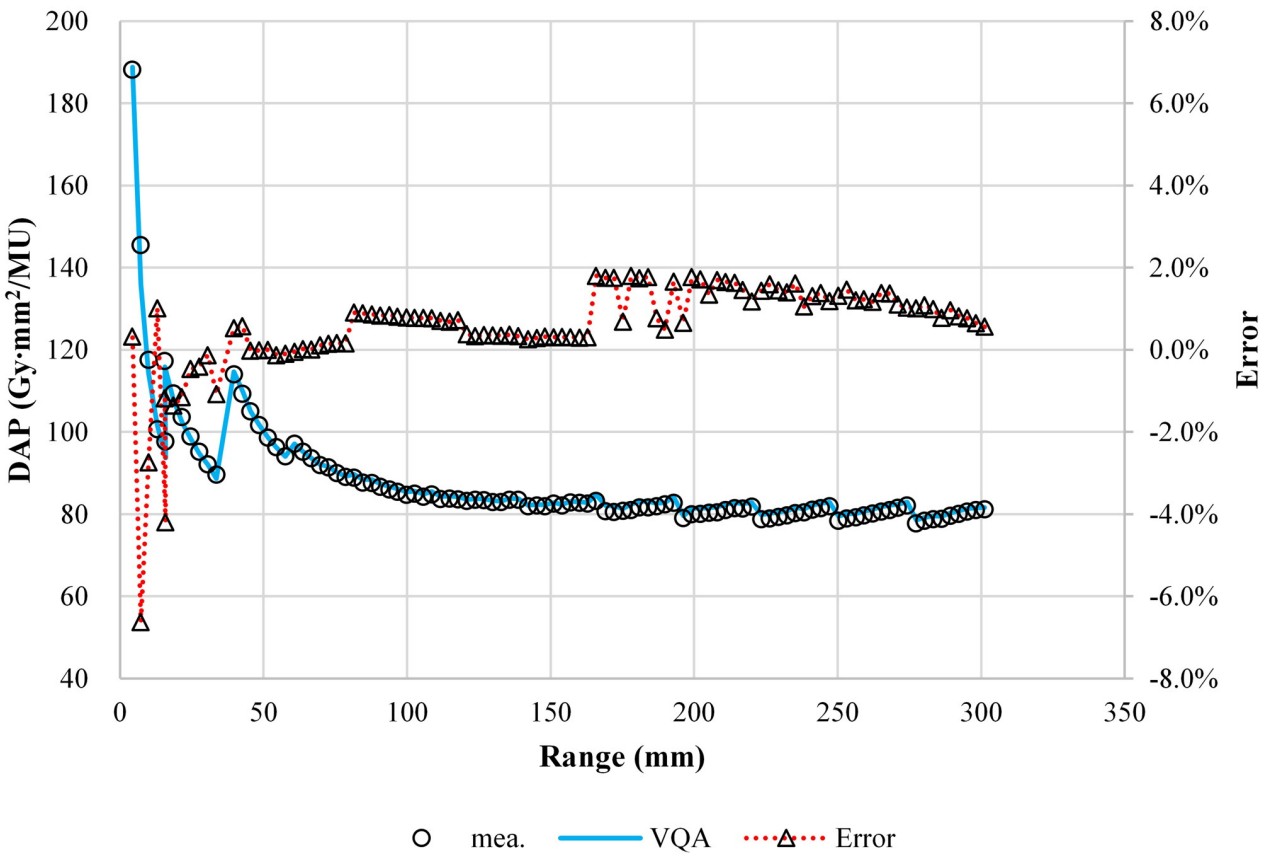

**Fig 5. Comparison of the absolute doses from the TPS and measurement in units of Gy mm²/MU for 100 energies.** The circles, solid lines, and dotted lines with triangles show the measured absolute dose, calculated absolute dose, and errors in the calculation relative to the measurement, respectively.

the region was rounded, indicating that the penumbral width was ~4.0 mm, and the width of the 95% dose was 6.0 mm shorter than the measured one [Fig 9(D) and Table 1]. In addition, the profile around the border between the acrylic and lung-equivalent materials was shallower than the measured profile.

## Validation of the biological dose calculation

We investigated the biological dose calculation accuracy by comparing the RBE values in the newly developed TPS with the RBE values calculated in a validated TPS that uses the same conceptual RBE model. Fig 10 compares the RBE values calculated using the newly developed TPS and the validated TPS; the difference is within 1.4% in the several irradiation conditions examined in this study.

## Measurements for patient-specific quality assurance

We performed measurements for PSQA after a new treatment plan developed with SFUD was approved. Fig 11(A) shows the measurement apparatus. Fig 11(B) shows a representative dose distribution from the TPS calculations, measurements, dose profile, and gamma index for a PC case and HNC case. Fig 11(C) shows that the averaged gamma passing rate was 99.0% and 98.4% for the PC and HNC cases, respectively, under gamma analysis using 3 mm/3%; notably,

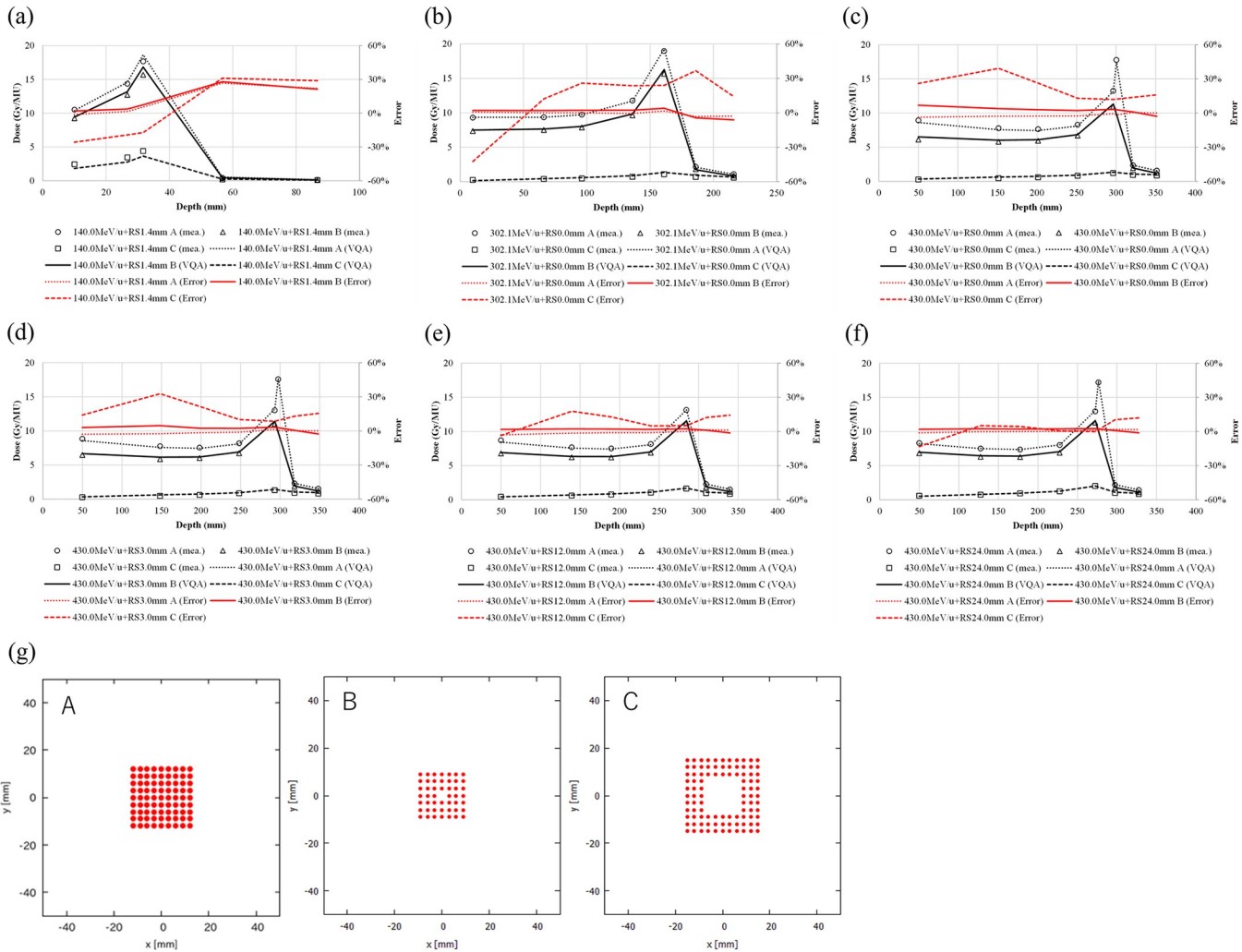

**Fig 6.** Doses measured in the irradiation of frame patterns A–C for a pencil beam. Point doses for a pencil beam with an energy of 430.0 MeV/u at energies (a–c) 140.0, 302.1, and 430.0 MeV/u, (a–c, respectively), and with an RS of 3.0, 12.0, and 24.0 mm (d–f, respectively). The measured absolute doses are shown for patterns A (circles), B (triangles), and C (squares), whereas the TPS-calculated absolute doses are shown for patterns A (dotted lines), B (solid lines), and C (dashed lines). The errors in the calculation relative to the measurement are shown for patterns A (red dotted lines), B (red solid lines), and C (red dashed lines). (g) Three selected frame patterns.

all cases were above 90%. The averaged gamma passing rate of the gamma analysis using 2 mm/3%, 2 mm/2%, and 1 mm/1% were 97.4%, 94.7%, and 74.8%, respectively, for the PC case and 97.4%, 93.1%, and 60.7%, respectively, for the HNC case.

## Discussion

In this study, we describe our comprehensive experience in commissioning the newly developed TPS for raster scanning of CIBs. Using the calibrated CT number-to-RSP curve, the modeled CIBs agreed well in water and provided us a better understanding of the limitations of the dose calculation accuracy in heterogeneous media. Between October 2018 and October 2020, we treated more than 900 patients with SFUD irradiated with fast-raster-scanned CIBs using the implemented model.

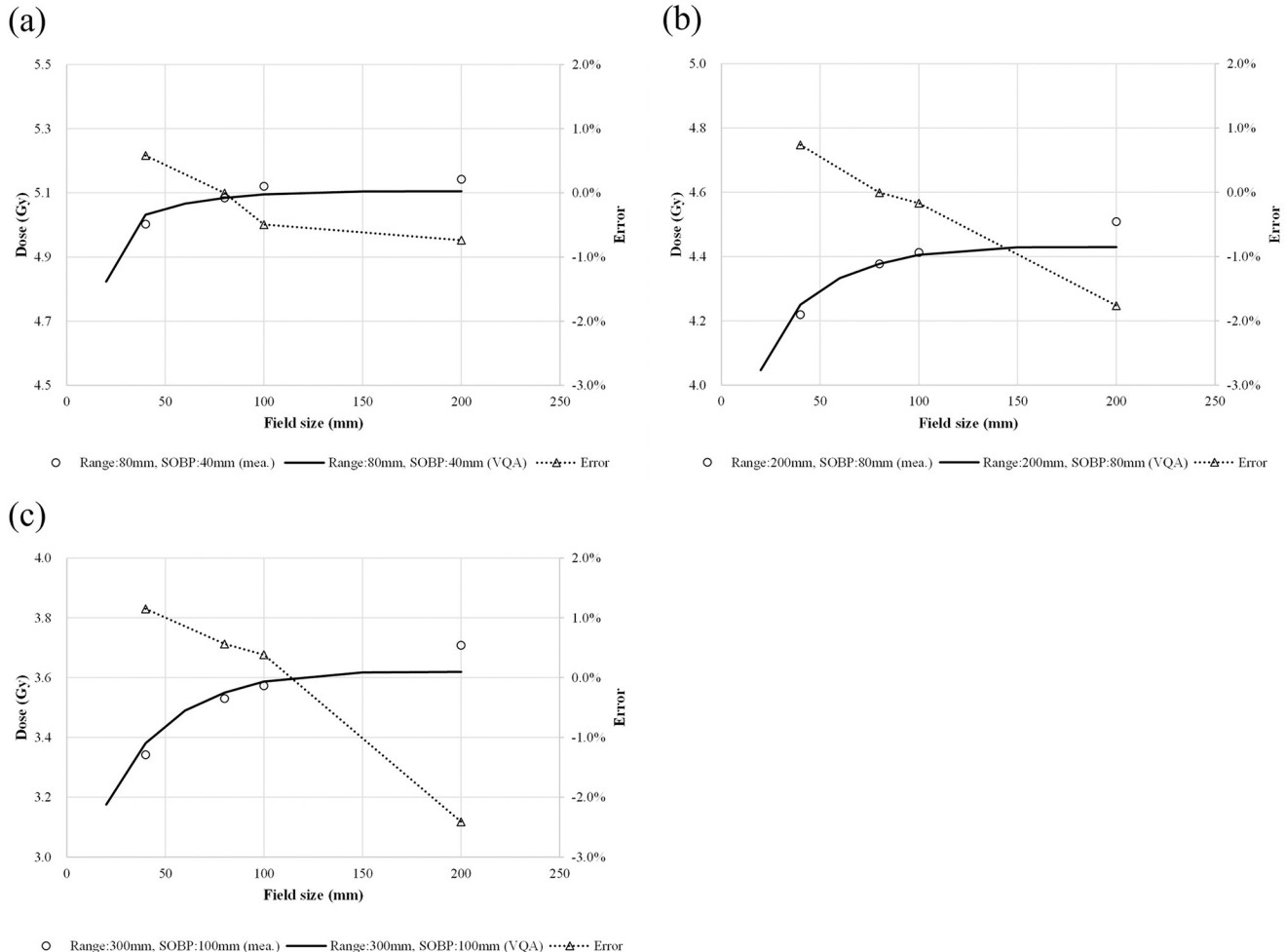

**Fig 7. Comparison between the TPS-calculated and -measured absolute point doses at the field center, and the SOBP for selected volumetric dose distributions as functions of field size ranging from 20 × 20 mm² to 200 × 200 mm².** The circles, solid lines, and dotted lines with triangles represent the measured absolute dose, calculated absolute dose, and errors in the calculation relative to the measurement (a) for an 80-mm range with a 40-mm SOBP, (b) for an 80-mm range with a 200-mm SOBP, and (c) for a 300-mm range with a 100-mm SOBP.

We determined the CT number-to-RSP curve using the stoichiometric calibration method, with representative tissues based on ICRP publication 110, and validated the accuracy by comparing the results with other curve calibration methods and RSP measurements. The representative tissues—corresponding to one reference adult phantom rather than including variations in age, physical status, and inter-individual differences—correspond to average tissue response. We adopted this calibration method because the polybinary tissue model requires a dedicated phantom. As shown in Fig 1(A), the CT number-to-RSP curve depends on the scanned subject's diameter because of the change in the CT scanner's beam quality. It would be suitable to create different curves contingent on the subject's size to reduce the uncertainty due to object size. Compared with the polybinary tissue model, our calibration curve showed a difference in high CT numbers, which typically correspond to the bone. This difference may have a greater or lesser impact not only on the calculation of the dose distribution but also on that of the range in a bony region. The experimental results agreed well with the RSP estimated from the calibrated curve to within the estimated uncertainty of the soft tissue (i.e., 1.4%).

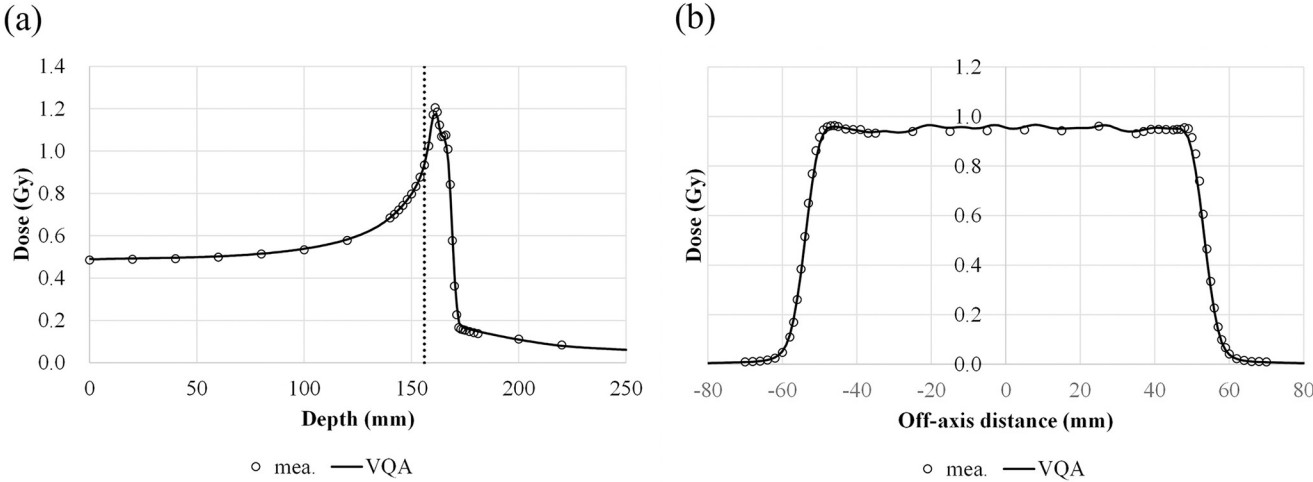

**Fig 8. Comparison of the TPS-calculated and measured dose profiles for a field size of 100 × 100 mm² with a 6-mm SOBP.** (a) The depth–dose profile along the central axis; the vertical dashed line represents the position of the measured lateral profile. (b) The lateral-dose profile at 156-mm depth; the circles and solid lines are the measured and calculated profiles, respectively.

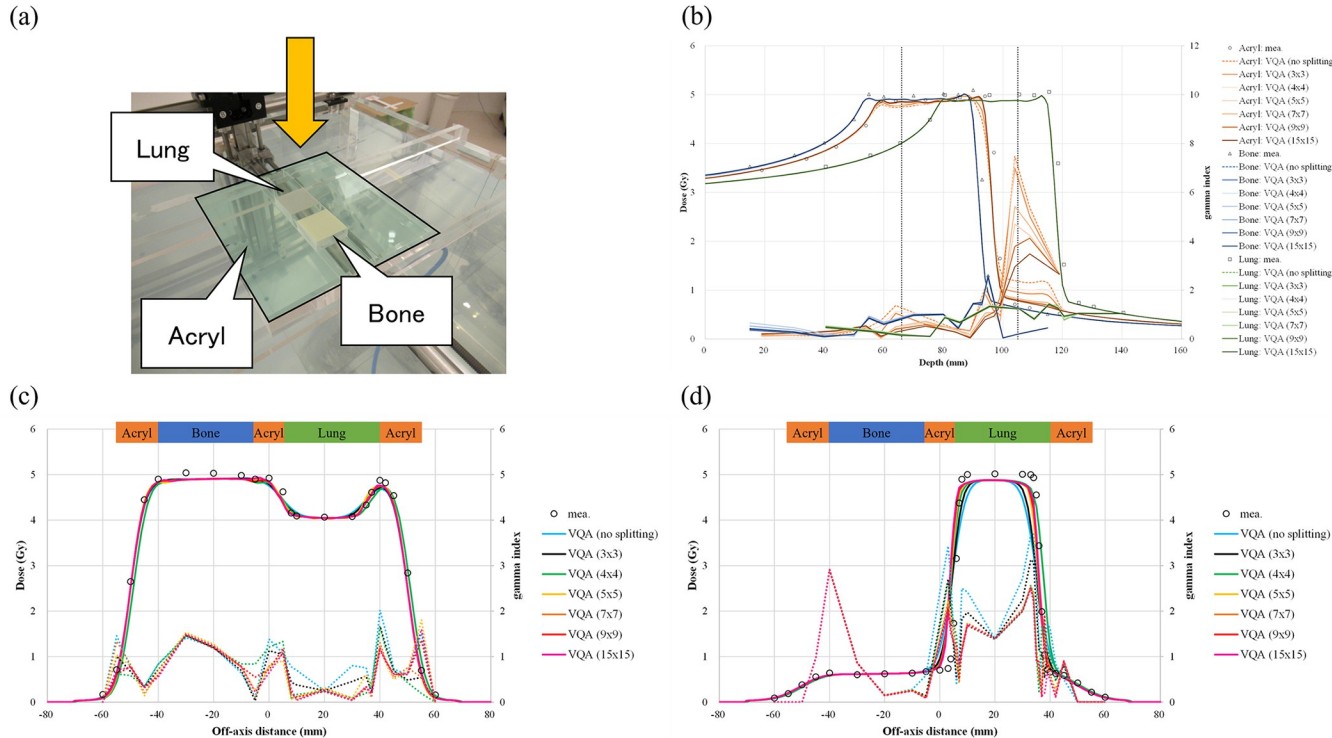

**Fig 9. Results of the scanned-beam model compared with measurements in heterogeneous media.** (a) The heterogeneous phantom and measurement setup; the yellow arrow represents the incident beam's direction. (b) The depth–dose profile penetrating each material; the circle, triangle, and square represent measurements in the acrylic, bone, and lung materials, respectively. The orange, blue, and green solid lines are the calculated profiles in acrylic, bone, and lung materials, respectively. The color in the legend of calculations changes according to the number of beam splittings. The vertical black dashed lines represent the measured lateral profile positions. (c and d) Lateral-dose profiles at 66 and 105 mm, respectively; the circles represent the measured profile, whereas the sky blue, blue, green, light yellow, yellow, orange, and pink solid lines are the calculated profiles with no splitting and with beam splittings 3 × 3 (default), 4 × 4, 5 × 5, 7 × 7, 9 × 9, and 15 × 15, respectively. The right vertical axis indicates the gamma value of 1D local-gamma-index analysis at a distance-to-agreement of 2 mm and a dose difference of 2%, with a 10% threshold.

**Table 1. Comparison of the dosimetric parameters of lateral-dose profiles penetrating the heterogeneous materials shown in Fig 9(C) and 9(D).**

| | 66 mm | | | | | | | | 105 mm | | | | | |
|---|---|---|---|---|---|---|---|---|---|---|---|---|---|---|
| | Penumbra 20%–80% | | Width of dose profile at | | | | | Dose difference at 0 mm | Penumbra 20%–80% | | Width of dose profile at | | | Dose differenceat 0 mm |
| | | | | 95% | | 50% | 5% | | | | 95% | 50% | 5% | |
| | Left | Right | Bone | Acrylic | Lung | | | | Left | Right | | | | |
| Splitting # | (mm) | | (mm) | | | | | | (mm) | | (mm) | | | |
| no splitting | −0.5 | 0.0 | −2.1 | 3.1 | −2.3 | −0.8 | 1.6 | −3.5% | 6.0 | 5.7 | −8.8 | 1.0 | −0.1 | 66.6% |
| 3 × 3* | −0.3 | 0.1 | −1.5 | 1.5 | −0.4 | −0.8 | 1.5 | −2.8% | 4.0 | 3.8 | −6.0 | 0.7 | −0.1 | 34.6% |
| 4 × 4 | −0.3 | 0.2 | −1.9 | 2.5 | −0.7 | −0.8 | 1.5 | −3.1% | 2.5 | 2.5 | −1.7 | 3.3 | −0.1 | 45.2% |
| 5 × 5 | −0.7 | −0.2 | −0.3 | −0.7 | 2.1 | −0.8 | 0.4 | −1.7% | 2.2 | 1.9 | −2.2 | 1.3 | −0.1 | 15.9% |
| 7 × 7 | −0.7 | −0.2 | −0.9 | 0.7 | 0.7 | −0.8 | 1.0 | −1.8% | 2.1 | 1.8 | −1.6 | 1.9 | −0.1 | 16.2% |
| 9 × 9 | −0.5 | −0.1 | −1.3 | 1.0 | 0.5 | −0.8 | 1.2 | −1.6% | 1.7 | 1.4 | −1.2 | 1.8 | −0.1 | 11.8% |
| 15 × 15 | −0.5 | 0.0 | −1.6 | 1.2 | 0.2 | −0.9 | 1.3 | −1.4% | 1.4 | 1.2 | −0.8 | 1.8 | −0.1 | 9.3% |

*The asterisk indicates the default splitting number in the TPS.

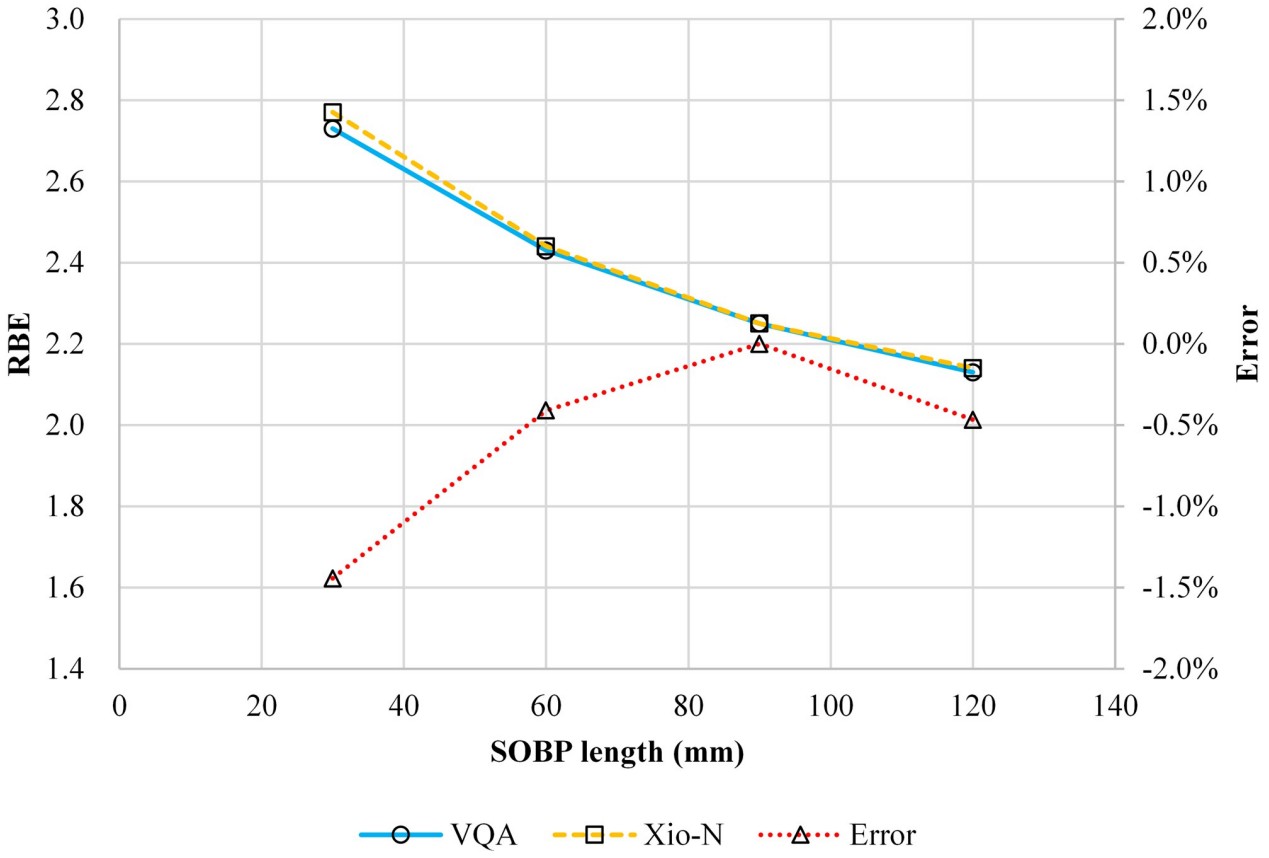

**Fig 10. RBE for the selected volumetric dose distribution of CIBs with a field size of 60 × 60 mm² and a range of 150 mm as a function of SOBP size from 30 to 120 mm.** The RBE values calculated using VQA Plan and Xio-N and the difference between the two are shown as a solid line with circles, dashed line with squares, and dotted line with triangles, respectively.

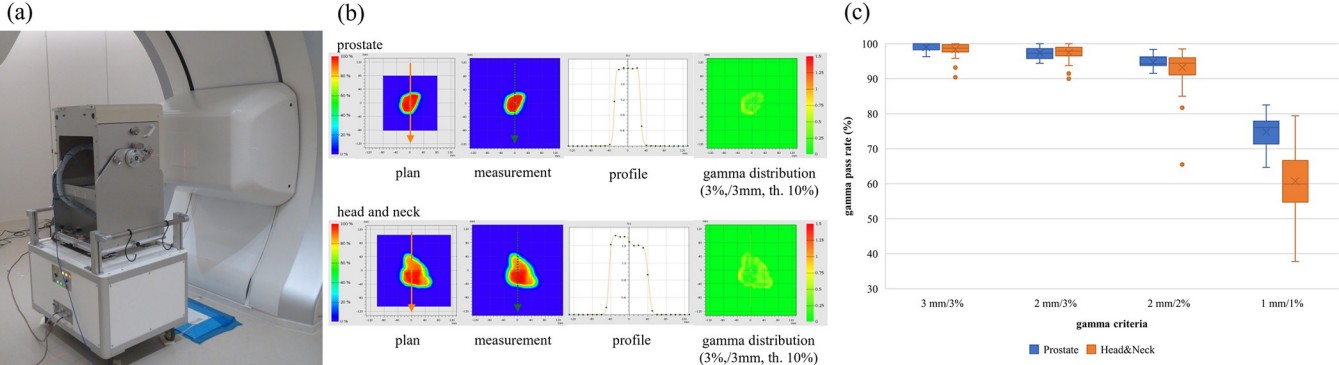

**Fig 11. PSQA performed in our institute.** (a) Measurement device combined with a 2D ionization chamber array and an accordion-type water phantom. (b) Typical results for PSQA. The upper and lower panels show the PC and HNC cases, respectively. The arrows on the dose distribution indicate the dose profile's direction. (c) The gamma passing rate in the clinical cases.

Besides, macroscopic heterogeneity due to air contamination may explain the difference between the RSP estimated from the calibrated curve and experiment [Fig 1(D) and 1(E)]. The uncertainties in the RSP were up to 2% for variations in human tissues [31]. In our stoichiometric calibration, we found the uncertainties in the RSP for the lung, soft tissue, and bone to be 6.6%, 1.4%, and 2.2%, respectively. In clinical practice, the range uncertainty is generally considered to be 3.5% in the treatment planning process to ensure target coverage. We also confirmed that this 3.5% range margin is sufficient to include the range uncertainty and uncertainty for each type of tissue, as estimated from our stoichiometric calibration.

In Fig 3, the values at the Bragg peak can change because of three competing effects: increased lateral scattering (causing a lower Bragg peak), decreased depth penetration (resulting in a high Bragg peak), and decreased nuclear fragmentation (causing a high Bragg peak). The small variability in the modeled data of the peak value may be due to statistical uncertainties (0.1%) in the MCSs used to model the IDDs. The heights of the Bragg peaks in the measurements were higher than those in the calculations, possibly because the measurement steps were smaller than the calculation resolution. In addition, the Bragg peak widths in the depth direction—which increases with the carbon-ion energy due to the range straggling—are reproduced in the modeled beam. The fragmentation tail was also found in the modeled beam. The increased dose in the fragmentation tail at high energies may be due to the increase in the amounts of nuclear fragmentation. The modeled beam reproduces these physical characteristics in the IDDs (Fig 3). The error in the absolute dose comparisons increased up to −6.6% at low energies of CIBs (Fig 5). A constant depth (e.g., 2 mm at 100 MeV/u) comes closer to the rising edge of the Bragg peak for the low-energy CIBs, making it difficult to measure the absolute dose accurately. In addition, the water phantom surface used in the measurements was aligned using a room laser, and matching the laser to the phantom surface may result in setup errors. There is a systematic jump in error at ~160-mm range, which could be due to the absolute correction factor used to adjust the absolute dose of CIB. The absolute correction factor was determined with the selected volume irradiations to minimize the dose difference between the calculated physical dose and measurement. As such, residual errors could be made. We adopted a triple Gaussian form for the beam model used to calculate the lateral-dose distribution. The first, second, and third components correspond to the primary carbon ions, particles scattered at small angles, and particles scattered at large angles, respectively. The profile in water (Fig 4) showed differences in the tail regions of the second and third components, whereas the profile close to the beam center agreed well because the results for in-air beam

profiles were good (Fig 2). The second and third components were modeled with patterned irradiation, and the maximum difference was up to −42.4% (Fig 6). We determined the modeling parameters for these components by fitting the data for all energies simultaneously and not by solving individual data [2]. Hence, the modeling parameters for each energy may include residual errors. These differences influence the TPS-calculated absolute dose for superimposed spots. Depending on the CIB energy, the absolute correction factor is considered to compensate for the difference between the calculated and measured absolute doses. The correction factor just involves multiplying each IDD to adjust it to the absolute dose [2] and ranges from 0.949–1.015. The correction factor worked well (Fig 7) for the beam model and beam-modeling process, including the MCSs and measurements, although it was not perfect. The dose distribution calculated with the modeled beam was acceptable for the superimposed spots in the water (Fig 8).

Splitting the beam into nine sub-beams revealed some differences between the TPS calculations and measurements. The method implemented in the TPS to correct for lateral heterogeneity is to split the beam only once at 700 mm upstream from the isocenter plane, whereas the original beam splitting algorithm divides a beam into sub-beams whenever the lateral heterogeneity is encountered in the beam path [5]. Hence, the beam size is too wide compared with the calculation grid (i.e., $2.0 \times 2.0 \times 2.0$ mm$^3$) to enable corrections to be made for lateral heterogeneity at the entrance to a heterogeneous phantom. For example, with a 21.0-mm RS, the main beam and sub-beam sizes at 272.8 MeV/u—the maximum energy used for volume irradiation—were 3.4 and 2.4 mm at the entrance, 3.6 and 2.5 mm at the isocenter, and 4.0 and 2.9 mm at 106 mm in the heterogeneous phantom, respectively. We performed an additional analysis with increased beam splitting numbers [Fig 9(B)–9(D) and Table 1]. As the beam splitting number increased, the profile from the TPS calculation approached the measurement. The local difference in the point dose between the TPS calculations and measurements at the center at 105-mm depth decreased to 9.3% with a beam splitting into 225 sub-beams ($15 \times 15$) from 34.6% with the beam split into nine sub-beams ($3 \times 3$), and the gamma index at the point decreased from 1.73 to 1.31 [Fig 9(D) and Table 1]. However, the difference to the dose in the SOBP was 1.3% and 5.0% with the 225 and 9 sub-beams, respectively. At 105-mm depth, the profile's shoulder became wider and the profile around the border became steeper, corresponding to a penumbral width of 2.6-mm shorter and 95% dose width 5.2-mm longer with the beam splitting into 225 sub-beams compared with splitting it into nine sub-beams [Fig 9 (D) and Table 1]. The increased beam splitting number improved the dose distribution penetrating a heterogeneous medium because the beam size became smaller at a given depth, although there was little improvement above a certain beam splitting number. This slowdown may be because the calculation's grid size was not small enough to accommodate the change in the divided beam's size. However, the increase in the beam splitting number substantially increased the computation time, which is inappropriate for clinical use. A study of the dose calculation models for a proton beam penetrating a heterogeneous medium showed that a ray casting model, an improved pencil-beam model considering off-axis heterogeneities with a single Gaussian form, exhibits a large error—10% or more—compared with MCSs when the inhomogeneities are far from the Bragg peak [3, 32], which is close to the geometry examined in this study. Thus, the observed difference in this study may be due to an intrinsic error in the dose calculation model. In terms of the absolute dose, the TPS yielded lower values than the measurements, probably due to the nonequivalence of water and body tissues, especially for nuclear reactions, which results in a different number of primary ions lost as well as different production rates of secondary particles. It was reported that a +4% difference was observed in IDDs for a 150-mm-thick layer of 40% $K_2HPO_4$ [8], which the authors used for emulating bone. This effect of the nonequivalence of water and body tissues intensifies for heavier ions

[33]. The dosimetric effect of the water nonequivalence in a patient may amount to 2.5% for extreme clinical cases in carbon-ion radiotherapy [34].

The biological dose calculation is crucial in the dose calculation for carbon-ion radiotherapy because it determines the absolute clinical dose and clinical dose distribution, which impact the absolute physical dose and physical dose distribution, respectively. We adopted the mixed beam model for the first time to calculate the biological dose for a scanning CIB. We confirmed that the accuracy of the newly developed TPS model was within 1.4% (Fig 10), which is acceptable for clinical use. The method used to validate the RBE calculation is limited as entails a mere comparison of the calculated RBE with the RBE measured in preclinical studies [2, 16, 28, 35]. Direct measurement of the LET [36] is another method for estimating the biological effect; however, it necessitates the use of a specialized detector, which is not yet feasible in a hospital. Validating the RBE values is crucial; however, it is beyond the scope of this study; we only propose a method for verifying RBE calculation accuracy using the RBE calculated by a validated TPS. As such, performing preclinical studies is unnecessary. This approach may not be useful for verifying RBE values calculated with other biological models but can be used for relative comparisons of RBE values calculated with different biological models. The mixed beam model is commonly used for biological dose calculations for a long time in our country, and it is thus the model used for the bulk of published clinical results for carbon-ion beams [17]. However, the MK model, which is a reformulation of the mixed beam model that incorporates the dose dependence of RBE, has started to be used in clinical treatments using carbon-ion radiotherapy [9]. Using the parameters of the commissioned beam, we performed an additional analysis to compare the clinical dose distributions based on the mixed beam with those obtained from the MK model, using a prototype VQA Plan that implemented the MK model. As shown in Fig 12, the clinical dose to the target was the same for both models,

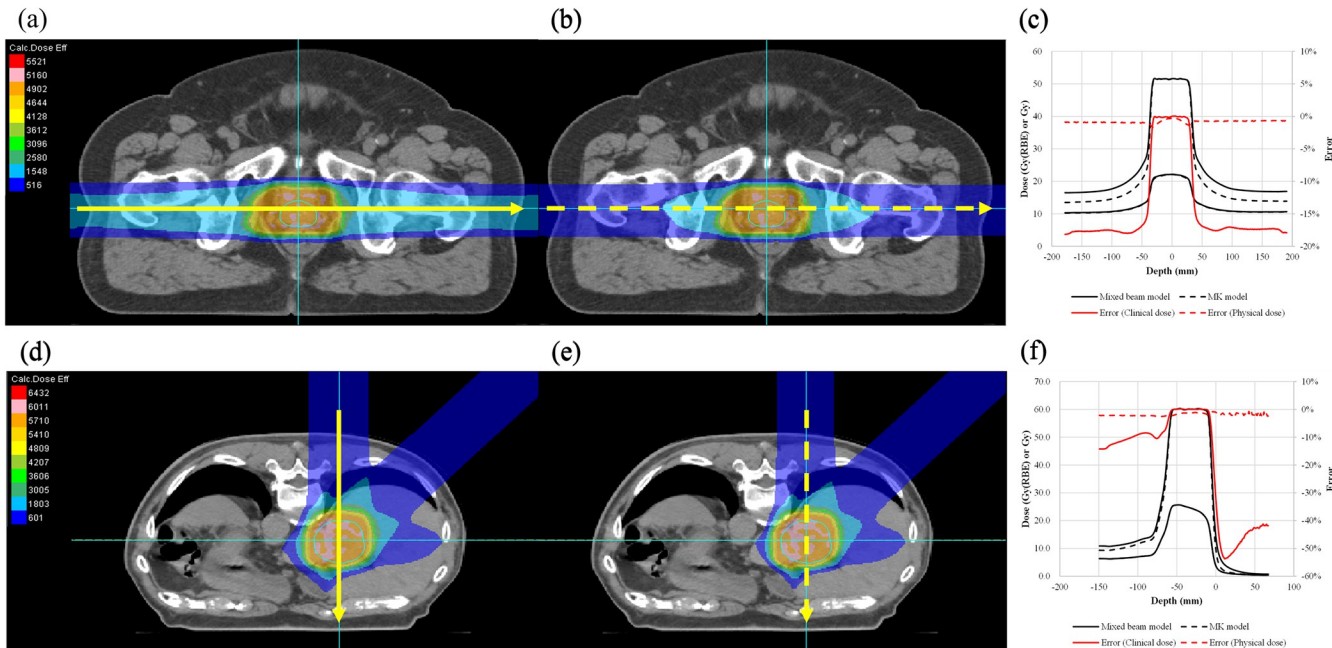

**Fig 12. Comparison of clinical and physical dose distributions from the mixed beam and MK models.** The PC case is prescribed to receive 51.6 Gy (RBE) in 12 fractions. This figure shows the dose distribution calculated with the (a) mixed beam model and (b) MK model. (c) The dose profiles, with differences between the RBE models, are shown by the yellow arrows. The liver case is prescribed to receive 60 Gy (RBE) in four fractions. The dose distributions shown are calculated with the (d) mixed beam model and (e) MK model. (f) The dose profiles with differences between the RBE models are shown by the yellow arrows.

whereas the dose at the entrance showed differences of up to 18% for the mixed beam model, depending on the prescribed dose level, with a smaller dose per fraction corresponding to a larger difference. This is because the MK model considers the dose dependence of the RBE using a CIB as the reference radiation.

The above PSQA results are clinically acceptable because the gamma passing rates were higher than the ones we determined for the PC and HNC cases [Fig 11(C)]. The PSQA results are also good for the strict criteria of the gamma analysis: 2 mm/2%. The 2D gamma analysis is influenced by uncertainty in the measurement depth. The sensitive volume of the 2D ionization chamber array is not particularly small ($0.06$ cm$^3$); hence, there is a possibility that the measurement and analysis were confounded by the volume effect of the array chambers [36]. Moreover, the lateral-dose gradient of the scanned CIB is not particularly sharp because of the lack of collimation, which is generally used in a broad beam method, whereas the dose distribution in the longitudinal direction has a steep gradient. Thus, the volume effect does not influence the lateral-dose distribution but only the longitudinal dose distribution. In addition, the 2D ionization chamber array may have been misaligned because of the setup errors of up to 1 mm because the accordion-type water phantom was aligned using the room laser, whereas errors due to changing the water thickness in the accordion-type water phantom are less than 0.1 mm, according to the specifications. These two factors—the volume effect and setup errors —can yield a difference in the point dose at the center of the 2D ionization chamber array. A 3D gamma analysis may mitigate the uncertainty in the measurement depth, as shown in [29]. Our normalization method using the PinPoint chamber may also mitigate this uncertainty because the small volume of the PinPoint chamber ($0.016$ cm$^3$) reduces the volume effect.

## Conclusions

In this study, we present our methods and experiences in commissioning a TPS for fast-raster scanning of CIBs. The TPS calculates physical doses using an analytical dose calculation algorithm, pencil-beam model with a triple Gaussian form, and beam splitting algorithm that uses the mixed beam model as the RBE model. To the best of our knowledge, these algorithms have never been implemented in an integrated TPS platform. The modeled beams reproduced both the physical and biological dose distributions in water. Analytic dose modeling for CIBs remains challenging, especially for heterogeneous conditions, because of complicated interactions with matter, such as multiple Coulomb scattering and nuclear interactions. We conclude that this TPS can be used clinically provided users understand the limitations of its accuracy for heterogeneous media.

## Supporting information

**S1 File.**
(PDF)

## Acknowledgments

The authors acknowledge and thank the staff in OHITC for their help with conducting measurements related to the commissioning, the staff in Osaka Heavy Ion Administration Company for their help in operating the accelerator in the commissioning, Mr. Kenji Matsuda (Hitachi, Ltd. Smart Life Business Management Division) for supporting data analysis, Dr. Takashi Akagi (Hyogo Ion Beam Medical Center) for providing the dedicated phantom for the polybinary tissue model, and the researchers in GHMC for the calculations of the RBE and QA team in the Japan carbon-ion radiation oncology study group and QA committee in OHITC

for a fruitful discussion on the commissioning. Finally, the authors would like to thank Enago (www.enago.jp) for the English language review.

## Author Contributions

**Conceptualization:** Masashi Yagi, Toshiro Tsubouchi, Noriaki Hamatani, Masaaki Takashina, Hiroyasu Maruo, Shinichiro Fujitaka, Hideaki Nihongi, Kazuhiko Ogawa, Tatsuaki Kanai.

**Data curation:** Masashi Yagi, Toshiro Tsubouchi, Noriaki Hamatani, Masaaki Takashina, Hiroyasu Maruo, Shinichiro Fujitaka, Hideaki Nihongi, Tatsuaki Kanai.

**Formal analysis:** Masashi Yagi, Toshiro Tsubouchi, Noriaki Hamatani, Masaaki Takashina, Shinichiro Fujitaka, Hideaki Nihongi.

**Funding acquisition:** Masashi Yagi.

**Investigation:** Masashi Yagi, Toshiro Tsubouchi, Noriaki Hamatani, Masaaki Takashina, Hiroyasu Maruo, Tatsuaki Kanai.

**Methodology:** Masashi Yagi, Toshiro Tsubouchi, Noriaki Hamatani, Masaaki Takashina, Tatsuaki Kanai.

**Software:** Masashi Yagi, Toshiro Tsubouchi, Noriaki Hamatani, Masaaki Takashina, Shinichiro Fujitaka, Hideaki Nihongi.

**Supervision:** Kazuhiko Ogawa, Tatsuaki Kanai.

**Validation:** Masashi Yagi.

**Visualization:** Masashi Yagi, Toshiro Tsubouchi, Noriaki Hamatani, Masaaki Takashina.

**Writing – original draft:** Masashi Yagi.

**Writing – review & editing:** Masashi Yagi, Toshiro Tsubouchi, Noriaki Hamatani, Masaaki Takashina, Hiroyasu Maruo, Shinichiro Fujitaka, Hideaki Nihongi, Kazuhiko Ogawa, Tatsuaki Kanai.

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
