## [Decision Letter · Decision Letter 0]

5 Nov 2021

PONE-D-21-22235Commissioning a newly developed treatment planning system, VQA Plan, for fast-raster scanning of carbon-ion beamsPLOS ONE

Dear Dr. Yagi,

Thank you for submitting your manuscript to PLOS ONE. After careful consideration, we feel that it has merit but does not fully meet PLOS ONE’s publication criteria as it currently stands. Therefore, we invite you to submit a revised version of the manuscript that addresses the points raised during the review process.

We look forward to receiving your revised manuscript.

Kind regards,

Aaron Specht

Academic Editor

PLOS ONE

Journal Requirements:

This work was partially supported by the JSPS KAKENHI 17K16437

This work was partially supported by the JSPS KAKENHI 17K16437

This work was partially supported by the JSPS KAKENHI 17K16437

Reviewers' comments:

Reviewer's Responses to Questions

**Comments to the Author**

1. Is the manuscript technically sound, and do the data support the conclusions?

Reviewer #1: Yes

Reviewer #2: Yes

Reviewer #3: Yes

Reviewer #4: Yes

2. Has the statistical analysis been performed appropriately and rigorously? 

Reviewer #1: Yes

Reviewer #2: Yes

Reviewer #3: Yes

Reviewer #4: Yes

3. Have the authors made all data underlying the findings in their manuscript fully available?

Reviewer #1: No

Reviewer #2: Yes

Reviewer #3: Yes

Reviewer #4: Yes

4. Is the manuscript presented in an intelligible fashion and written in standard English?

Reviewer #1: Yes

Reviewer #2: Yes

Reviewer #3: Yes

Reviewer #4: Yes

5. Review Comments to the Author

Reviewer #1: Review comments of PONE-D-21-22235

Commissioning a newly developed treatment planning system, VQA Plan, for fast raster scanning of carbon-ion beams

General comments:

The readability is acceptable in general with minimal grammar errors. It does come across a little dense at times. Although the manuscript is reasonably well readable, the English could be improved.

1. Paper structure and presentation are too lengthy in the current version and need to be more concise.

2. Tables and figures, especially the quality of all figures, need to be improved in the revision.

3. Language also needs to be improved by native English speakers. Please consult the reviewers' comments for more information.

The manuscript is lacking of the core value introduction of VQA which is a flaw that detracts from the positive contributions of this article. The author is encouraged to address more superiority of their planning system VQA.

Specific comments please refer to the attached file in reviewer's comments.

Reviewer #2: Thank you very much for choosing me to review this article.

I have read this article in its entirety three times, and I believed that this research is one of the most complete studies I have ever seen in this field, and in many cases the scientific principles governing the subject, both in the fields of physics and radiobiology. However, considering the title and goal of research which focused on commissioning of a new treatment planning system (TPS) in the field of carbon therapy with considerations of all of physical and radiobiological aspects, there is a vague point for me. In this research, the LQM model of cell survive are used for indication of Salivary Gland Tumor (SGT) cells respond to Carbon Ions irradiation. Despite of Target Model of cell survive, the LQM model, based on various studies, to justify the behavior of cells in the face of low-LET electromagnetic radiation, such as X-rays and gamma rays, whose non-particle properties are predominant in dealing with different cells and tissues. However, in the field of particle therapy such as carbon therapy, the primary radiation particles, namely carbon and subsequent products are mainly of different isotopes and fragments, all of which are characterized by their particle properties and the contribution of non-particle electromagnetic radiations in this process are very small. Although the particle and non-particle radiations (photons and Ions) contribution considered in the equation 3, however the effect of each part of products (Isotopes, Fragments, Photons …) on SGT survival curve and correspondent Alpha and Beta are not presented clearly. It seems that for calculation of biological dose for each voxel (dbio.i), the weight of effects of all types of radiations (Photons and Ions…) on final cell Survival Curve shape and then the correspondent Alpha and Beta must be considered. So, in order to introducing an equation for calculation of biological dose in each voxel and uses in new TPS algorithm, the three important topics must be considered:

1- Demonstration some documents which confirmed that the LQM model is suitable for mixed beam radiations in the field of SGT carbon therapy

2- Demonstration of the effect and contribution of mixed beam radiation products (Photons and particles) on SGT cell survival curve quantitatively

3- Demonstration of amounts of the Alpha and Beta components related to each product of mixed beam radiation separately

Your Sincerely. Dr A.Zeinali

Associate Prof. of Medical Physics, UMSU.IR.Iran.

Reviewer #3: This is a very well written and comprehensive paper. As far as the commissioning process goes, there are only limited other tests that I would like to see before implementing a new TPS. The figures need some improvement and I think further patient-specific QA should be included as a way to address limitations of the TPS, but other than that I only have minor comments.

Major comments:

Figures: I would recommend colorizing many of the figures (especially Figs. 3, 4, 5, and 10) to make lines stand out from each other. Given that the manuscript is currently very long, I would also say Fig. 7 can be removed and accompanying discussion can be reduced without greatly impacting the manuscript. For Fig. 3, I would recommend greatly reducing the number of IDDs plotted (plotting every 5-10 MeV/u would be sufficient). Currently, there is too much data on the plot to reasonably separate curves.

Beam splitting and lateral heterogeneities: the authors mention in the introduction that the main difference between VQA and other treatment planning systems is handling of lateral heterogeneities and biological dose calculation. I would argue then that more discussion of lateral heterogeneities, specifically when errors arise as in Fig. 9, should be included. There is some very insightful discussion about beam splitting and its impacts on errors, but more concrete guidance should be provided as this paper should fully describe the limitations of the dose calculation. Also, errors of both 34% and 9% (Fig. 9) seem clinically unacceptable, but do appear to be low as a percentage of prescription dose. A 1-D gamma analysis may be more useful here than direct dose comparison. It is mentioned that the increase in computational time makes extra beam splitting clinically inappropriate, but I would argue that 34% disagreement is also clinically inappropriate. Are there certain clinical scenarios where some degree of beam splitting should be used in the calculation? Please comment on this in the discussion.

Since this is a commissioning paper, tighter gamma tolerances should be shown (2%/2mm and 1%/1mm) in addition to the current data per AAPM TG-218 (Miften, Moyed, et al. "Tolerance limits and methodologies for IMRT measurement‐based verification QA: recommendations of AAPM Task Group No. 218." Medical physics 45.4 (2018): e53-e83) to investigate where the model may break down.

Minor comments:

Methods:

Is live rastering of the scanning beam modeled in other treatment planning systems mentioned in the introduction? If not, what is the estimated dosimetric impact of not modelling this rastering effect?

P9L187: “the first component” is this referring to a component of one of the above equations? Please specify.

P15L333: Is this supposed to be “except for accelerator energies below 100 MeV/u”?

A graphical explanation or figure showing the three selected frame patterns (P16, L347) would be very helpful

P17L390-392: “We measured the RSP for each material comprising the phantom in the same manner…and this was reflected in the calculation by overwriting the CT-number function after inversely estimating the CT number of the measure RSP in the table because phantom material was not real tissue and CT number acquired from CT image was not as accurate as that of the tissue”… If you override the CT numbers during the validation, aren’t you negating the value of the validation process? Could the authors please explain further?

Results:

Fig. 1 and later discussions: It seems like there is large error around 1000 HU (inset c), a value which can be seen in dense bone in vivo. Will this larger error be reflected in robust analyses for patient plans? In the methods section (L473), the authors state a range margin of 3.5% is enough to include range uncertainty in composite range calculations. Is this true even when treating sites with lots of bone (for example, in the sacrum)? Also, the right-hand side axis on insets (b) and (c) do not line up with the grid lines.

Fig 5 – there is a systematic jump in error at about 160 mm range, why? A known error in absolute output around 2% seems rather high for clinical use, so the cause of this should be discussed.

For Fig. 6 and other figures showing errors, it may be better to show error in absolute dose rather than relative dose. Again, the right-hand y-axis does not line up with the grid lines on the plot.

Table 1: should “Dose at _ mm” columns be dose difference?

Discussion:

P31L678: what are statistical uncertainties of the MC simulation?

P35L766: It’s not that RBE validation is not necessary, rather that it is not feasible and this is an acceptable alternative. RBE validation is something that is very much necessary within the field, but it is fair to say it is beyond the scope of this manuscript.

Reviewer #4: The article is written well. however I find it bit lengthy. Authors may consider shortening where possible.

Some figures in the manuscript are of poor quality to interpret.

Authors may consider mentioning what DTA and DD were used in the gamma analysis.

6. PLOS authors have the option to publish the peer review history of their article (what does this mean?). If published, this will include your full peer review and any attached files.

Reviewer #1: **Yes: **Jia-Ming Wu, PhD

Reviewer #2: **Yes: **Ahad Zeinali

Reviewer #3: **Yes: **Joseph Harms

Reviewer #4: **Yes: **Jothybasu Selvaraj

---

## [Author Response · Author response to Decision Letter 0]

2 Mar 2022

We attached response to reviewers as separated files. Please see at the attached files.

---

## [Decision Letter · Decision Letter 1]

22 Apr 2022

Commissioning a newly developed treatment planning system, VQA Plan, for fast-raster scanning of carbon-ion beams

PONE-D-21-22235R1

Dear Dr. Yagi,

We’re pleased to inform you that your manuscript has been judged scientifically suitable for publication and will be formally accepted for publication once it meets all outstanding technical requirements.

Kind regards,

Aaron Specht

Academic Editor

PLOS ONE

Additional Editor Comments (optional):

Reviewers' comments:

Reviewer's Responses to Questions

**Comments to the Author**

1. If the authors have adequately addressed your comments raised in a previous round of review and you feel that this manuscript is now acceptable for publication, you may indicate that here to bypass the “Comments to the Author” section, enter your conflict of interest statement in the “Confidential to Editor” section, and submit your "Accept" recommendation.

Reviewer #3: All comments have been addressed

Reviewer #4: All comments have been addressed

2. Is the manuscript technically sound, and do the data support the conclusions?

Reviewer #3: (No Response)

Reviewer #4: Yes

3. Has the statistical analysis been performed appropriately and rigorously? 

Reviewer #3: (No Response)

Reviewer #4: Yes

4. Have the authors made all data underlying the findings in their manuscript fully available?

Reviewer #3: (No Response)

Reviewer #4: No

5. Is the manuscript presented in an intelligible fashion and written in standard English?

Reviewer #3: (No Response)

Reviewer #4: Yes

6. Review Comments to the Author

Reviewer #3: (No Response)

Reviewer #4: The authors have addressed all my comments. The authors have justified the length of the manuscript.

7. PLOS authors have the option to publish the peer review history of their article (what does this mean?). If published, this will include your full peer review and any attached files.

Reviewer #3: **Yes: **Joseph Harms, PhD

Reviewer #4: **Yes: **Jothy Selvaraj

---

## [Editor Report · Acceptance letter]

29 Apr 2022

PONE-D-21-22235R1 

Commissioning a newly developed treatment planning system, VQA Plan, for fast-raster scanning of carbon-ion beams 

Dear Dr. Yagi:

I'm pleased to inform you that your manuscript has been deemed suitable for publication in PLOS ONE. Congratulations! Your manuscript is now with our production department. 

Kind regards, 

on behalf of

Dr. Aaron Specht 

Academic Editor

PLOS ONE